# Training Dynamics of the Cooldown Stage in Warmup-Stable-Decay Learning Rate Scheduler

**Aleksandr Dremov, Alexander Hägele, Atli Kosson, Martin Jaggi**
*EPFL, Switzerland*

**Reviewed on OpenReview:** *https://openreview.net/forum?id=ZnSYEcZod3*

## Abstract

Learning rate scheduling is essential in transformer training, where the final annealing plays a crucial role in getting the best performance. However, the mechanisms behind this *cooldown phase*, with its characteristic drop in loss, remain poorly understood. To address this, we provide a comprehensive analysis focusing solely on the cooldown phase in the Warmup-Stable-Decay (WSD) learning rate scheduler. Our analysis reveals that different cooldown shapes reveal a fundamental *bias-variance trade-off* in the resulting models, with shapes that balance exploration and exploitation consistently outperforming alternatives. Similarly, we find substantial performance variations — comparable to those from cooldown shape selection — when tuning AdamW hyperparameters. Notably, we observe consistent improvements with higher values of $\beta_2$ during cooldown. From a loss landscape perspective, we provide visualizations of the landscape during cooldown, supporting the *river valley* loss perspective empirically. These findings offer practical recommendations for configuring the WSD scheduler in transformer training, emphasizing the importance of optimizing the cooldown phase alongside traditional hyperparameter tuning.

## 1 Introduction

Learning rate scheduling remains a critical component in training transformer-based language models. Since the introduction of transformers by Vaswani et al. (2017), learning rate scheduling has become standard practice in transformer training pipelines and remains prevalent in recent works such as Meta AI (2024) and DeepSeek-AI et. al (2024). Among various proposed strategies, cosine decay (Loshchilov & Hutter, 2017) remains one of the most commonly employed methods.

An alternative approach, the *Warmup-Stable-Decay (WSD)* scheduler, has recently gained popularity. This method features three phases: an initial warmup phase with a linear learning rate increase, a stable phase maintaining constant rates, and a final decay phase reducing rates to zero. Hägele et al. (2024); Hu et al. (2024) demonstrated that WSD achieves performance comparable to, or better than, cosine decay in transformer-based language modeling. Importantly, it allows training without a predefined training length and enables continuation from any point in the stable phase.

Notably, WSD-trained models exhibit loss curves that remain relatively high during the stable phase but drop sharply during cooldown. Previous work has provided extensive analysis of WSD parameters and their impact on final performance (Hägele et al., 2024) as well as the role of the warmup phase (Kosson et al., 2024b). However, the cooldown phase dynamics remain under-explored.

This work aims to address this gap by investigating the behavior and impact of the cooldown stage in the WSD learning rate scheduler, providing insights into its effects. We propose a bias-variance framework for cooldown learning rate scheduling shapes, through which we explain why some shapes tend to perform better than others. The schematics for the proposed framework are shown in Figure 1. The study examines hyperparameter effects, including AdamW's beta parameters, weight decay, and batch size configurations, with experimental results verifying alignment between observed outcomes and the proposed framework.

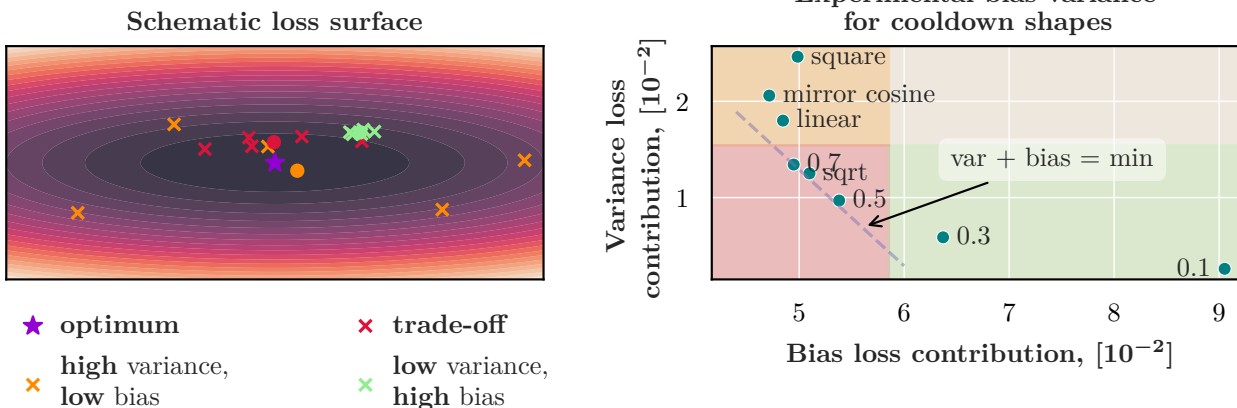

Figure 1: **Conceptual visualization of the relationship between different learning rate cooldown shapes and their bias and variance. On the left**, we show a conceptual visualization of loss surface points in two dimensions. Colors refer to different approaches, with crosses denoting the result of independent runs and the circles marking the mean for each method. The methods exhibit different *bias-variance* trade-offs with respect to the optimum in the middle, where the *variance* measures the spread in the individual solutions and the *bias* is the error in the mean solution. **On the right**, we present different cooldown shapes based on their bias-variance characteristics. Here the bias is computed as the loss difference between the average model of experiments with different data orderings (shuffling) and a better reference model (obtained by longer training on the same data), whereas the variance is the difference between the average loss of experiments and the loss of the average model of these experiments. Coordinates can be connected to the left plot where bias can be understood as the loss distance between the circle dot (cluster mean) and a star (optimum). Similarly, variance is the difference between the average loss for individual models (crosses) and the loss of the mean model (circle).

## 2 Background: Learning Rate Scheduler

Learning rate scheduling is a technique used in neural network training where the optimization algorithm's learning rate is adjusted during training according to a predefined rule. Figure 2 illustrates various scheduling techniques.

A warmup phase is commonly employed at the beginning of training, during which the learning rate is gradually increased from small values to the desired level. This approach allows the model to tolerate a higher peak learning rate during training, improves stability in the initial phase, and positively affects final performance (Kosson et al., 2024b; Kalra & Barkeshli, 2024).

Cosine learning rate scheduling is a widely used technique. This method adjusts the learning rate following a cosine curve, starting at a maximum value after a warmup period and gradually decreasing it to a minimum value, often near zero (Bergsma

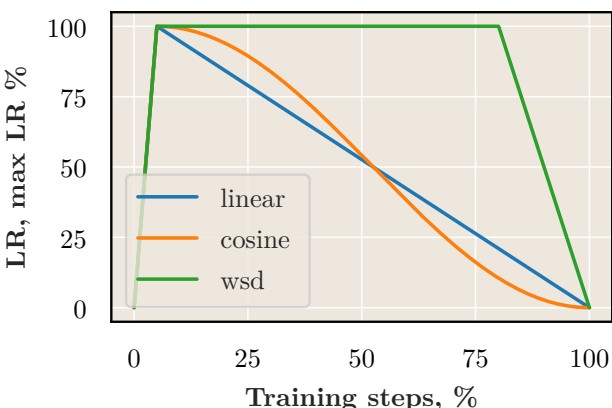

Figure 2: **Different learning rate schedules.** We show the cosine, WSD, and linear schedules. A warmup stage of 5% of the training length is added to each schedule.

et al., 2025). The key parameter for the cosine schedule is the training duration. As observed in Hoffmann et al. (2022), achieving optimal model performance for a specific token count requires matching the training

duration to the cosine scheduler length. This finding was also verified by Hägele et al. (2024). Therefore, cosine learning rate scheduling has a significant drawback: training sessions resumed after the cosine curve ends result in worse performance compared to training sessions with a cosine schedule of the required length.

### 2.1 Warmup Stable Decay Scheduler

An alternative to the cosine learning rate scheduler is the warmup-stable-decay (WSD) scheduler (Figure 2). It consists of three stages: warmup, a constant learning rate, and decay (cooldown). The cooldown stage is relatively short, typically around 20% of the total training duration. A notable feature of WSD is that it avoids the primary drawback of the cosine scheduler. Training can be resumed seamlessly from any part of the constant learning rate stage without performance degradation.

The performance of this scheduler for language modeling, particularly in the context of the Llama-like model (Touvron et al., 2023), is analyzed in depth by Hägele et al. (2024). Their key finding is that the WSD learning rate scheduler can match or even surpass the performance of the cosine scheduler.

A critical part of WSD performance is the cooldown stage. As seen in Figure 3, a dramatic drop in validation perplexity occurs during the cooldown stage.

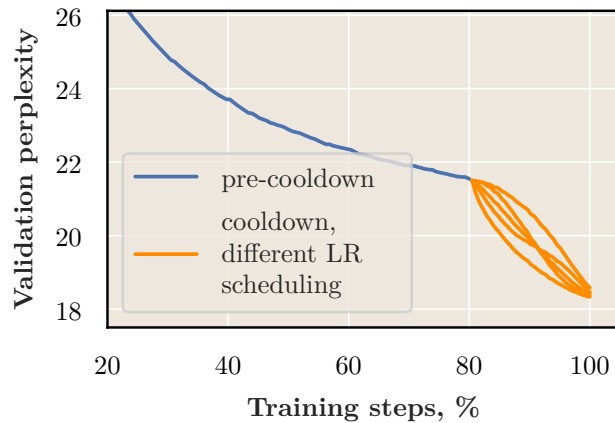

Figure 3: **The drop in perplexity during the cooldown stage.** The cooldown stage begins at 80% of the training, which leads to a dramatic drop in perplexity during the final 20%.

Such performance improvement is also observed in downstream tasks, as demonstrated by Hägele et al. (2024). Gaining insights into the cooldown stage may reveal ways to further improve performance. For this reason, this work focuses on the specifics of the cooldown stage.

## 3 Experimental Setup

As noted, this work focuses exclusively on the cooldown stage. Therefore, the pre-cooldown model is kept consistent across all experiments, and only the hyperparameters during the cooldown stage are modified, unless otherwise noted.

The model used is a standard decoder-only transformer with 210 million parameters (Vaswani et al., 2017), identical to Llama (Touvron et al., 2023). Unless otherwise specified, we use the AdamW optimizer with weight decay (Loshchilov & Hutter, 2019; Kingma & Ba, 2017) and standard LLM training parameters. Training is conducted on a subset of SlimPajama (Soboleva et al., 2023). The dataset is split into training and validation parts, and validation perplexity is used to evaluate performance. We provide a detailed description of the model architecture and used techniques in App. A.

The pre-cooldown model is trained for 26,400 iterations (2.7B tokens) with 300 warmup steps. The cooldown stage length is commonly set to 20%, which has been identified as an effective fraction for achieving strong final performance, as shown by Hägele et al. (2024). During the cooldown stage, the learning rate is always reduced to zero.

## 4 Cooldown Shape: Does It Matter?

As noted by Hägele et al. (2024), the shape of the cooldown phase significantly impacts the performance of WSD. Figure 4 illustrates the validation perplexity for various cooldown shapes, as well as the shapes them-

selves. Interestingly, validation perplexity closely follows the shape of the learning rate and reaches different final values. However, Hägele et al. (2024) did not explore the reasons behind the different performance observed for different cooldown shapes.

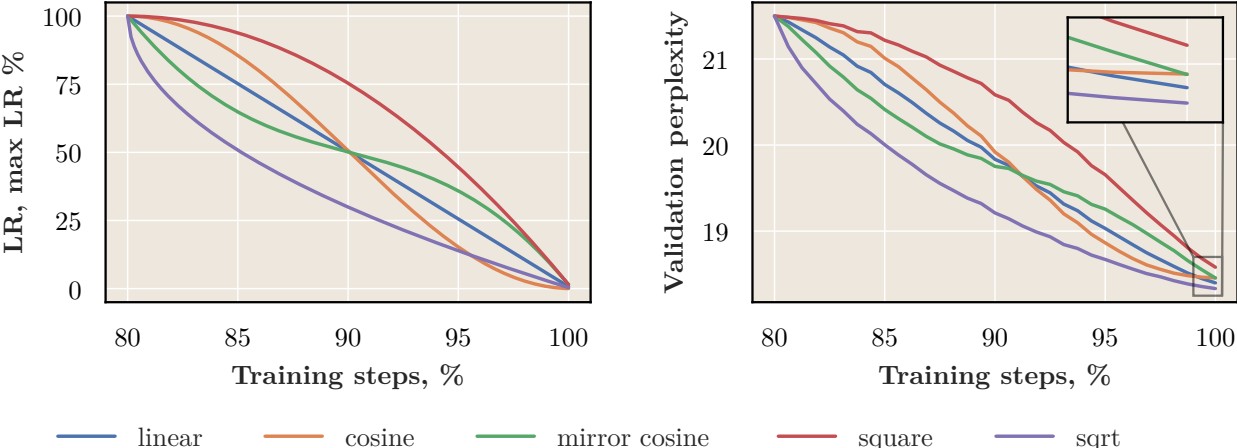

Figure 4: **The surprising effect of cooldown shapes on final performance. On the left,** different shapes of learning rate scheduling during the cooldown stage. **On the right,** the corresponding validation perplexities. Validation perplexity follows a pattern similar to that of the learning rate, but the final performance varies. It is clear that cooldown shapes affect the course of training as well as the final perplexity.

### 4.1 Bias-Variance Point of View

To understand why different cooldown shapes yield varying performance, we examine the underlying optimization dynamics. Specifically, we propose the following: different cooldown shapes create a fundamental bias-variance trade-off that explains their varying performance. Figure 1 illustrates this concept schematically. Essentially, the cooldown shape controls a critical balance: more aggressive exploration (high learning rates) of the loss surface increases variance across models (if trained with different data orderings), while potentially yielding superior performance; in contrast, focusing on exploiting (low learning rates) the current region of the loss surface produces more consistent solutions but may sacrifice overall quality.

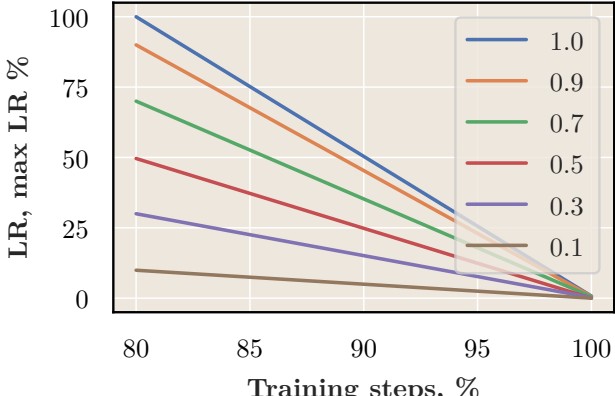

Figure 5: **Lowered linear cooldown shapes.** The parameter controls the starting learning rate, where 100% is the learning rate of a constant stage.

To investigate this, we propose the following framework. For each cooldown shape, we train $N$ models from the same constant stage model with identical hyperparameters but different data orders. Each model is trained on a random permutation of the same dataset portion for 33,000 steps in total (3.4B tokens). In addition to the cooldown shapes mentioned earlier, we use the "lowered linear" shapes family displayed in Figure 5.

We train models with a much longer cooldown stage (59,400 steps in total, 6B tokens) using the *sqrt* cooldown shape, which represents a good solution in the optimization landscape and is therefore referred to as *reference models*. The dataset portion used for the reference models is the same as that for the shorter experiments,

but is permuted (with repetitions) to match the extended training duration. In total, $N = 9$ models are trained for different data permutations.

For each cooldown shape, we calculate two metrics: first, the performance on the validation dataset of the average model (in weight space) over different data permutations; and second, the performance of each distinct experiment (specific data permutation). The same procedure is applied to the reference models. In other words, we obtain the following: the performance of each specific cooldown shape for all data permutations, the performance of the cooldown shape of the averaged model across different data permutations, and the averaged reference model performance.

Formally, let $m_i$ be the model with a fixed cooldown shape and the $i$-th data permutation, $m^*$ be the average of reference models (in weight space), and $L(\cdot)$ the validation loss of a specified model. With this notation, we can formulate the bias-variance decomposition of model performance in relation to a better model $m^*$:

$$\mathbb{E}_i\left[L(m_i) - L(m^*)\right] = \mathbb{E}_i\left[L(m_i)\right] - L(m^*) = \underbrace{L(\mathbb{E}_i\left[m_i\right]) - L(m^*)}_{\text{bias}} + \underbrace{\mathbb{E}_i\left[L(m_i)\right] - L(\mathbb{E}_i\left[m_i\right])}_{\text{variance}}, \qquad (1)$$

where $\mathbb{E}_i$ is the expectation with respect to the different data permutations. Concretely, $\mathbb{E}_i\left[m_i\right]$ is the average model in weight space across different data permutations, and $\mathbb{E}_i\left[L(m_i)\right]$ is the average loss of different models across different data permutations. We believe that the proposed framework captures information about optimization stability through the variance of performance across different experiments, and captures the distance to a better solution achievable by learning on the same data. Surprisingly, similar results to those in the following can be achieved in weight space. We show how different measures of the bias and variance affect the results in App. C.

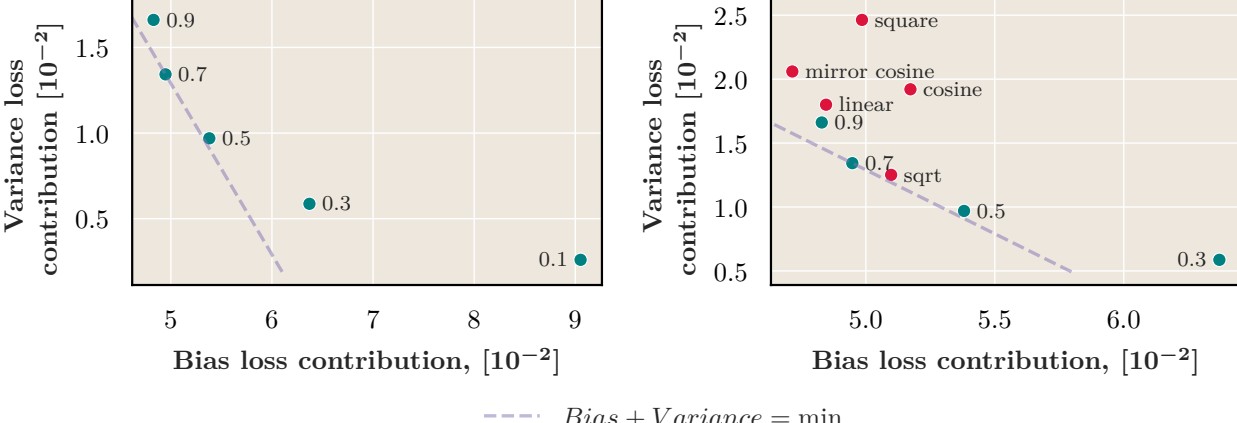

Figure 6: **Bias-variance plot for different cooldown shapes.** We measure *bias* as the difference in validation loss of the reference model (average of models trained for longer on the same data but different permutations) and the loss of the average of experimental models (different data permutations). *Variance* is the difference between the average loss of experiments and the loss of the average model of these experiments (Equation 1). We also display a line where the minimum *bias + variance* for experiments is achieved. **On the left,** we compare only lowered linear shapes (Figure 5), which exhibit an expected decrease in variance and increase in bias with decreasing parameter value. **On the right,** the full range of nonlinear shapes along with selected lowered linear shapes for reference. The *lowered linear* shape with parameter 0.7 and the *sqrt* shape occupy favorable positions, achieving a balanced trade-off between bias and variance.

**Plot interpretation.** The experimental results are presented in Figure 6. From the figure, we observe a clear relationship between bias and variance, which we believe explains why some cooldown shapes tend to perform better than others. In this setting, the optimal cooldown shape is one that achieves the minimum of *bias + variance*. We note that the *lowered linear* shape with parameter 0.7 and the *sqrt* shape occupy an optimal position, achieving a balance between bias and variance. This can be interpreted from an exploration/exploitation perspective: shapes with *high variance and low bias* tend to explore the loss surface more extensively, leading to significantly different solutions. Conversely, shapes with *low variance and high bias* tend to exploit the current solution by descending into the loss basin.

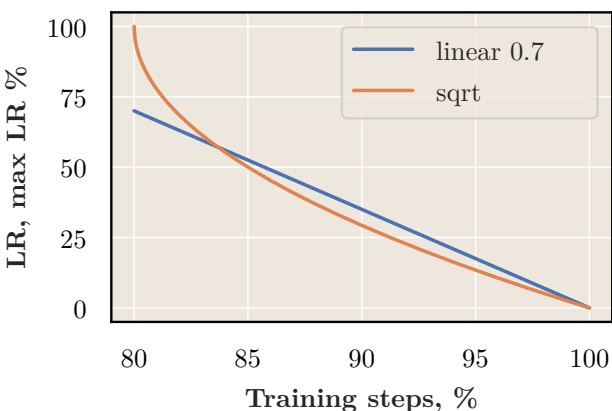

Figure 7: **Comparison of the *sqrt* cooldown shape and *lowered linear 0.7*.** The shapes are similar, achieving comparable performance, although the *lowered linear 0.7* exhibits slightly better perplexity.

One notable feature of Figure 6 is that *sqrt* lies close to the lowered linear shape with a parameter of 0.7. We observe that the lowered linear 0.7 cooldown achieves performance comparable to, or even better than, *sqrt*. This result shows that there is nothing special about the shape of *sqrt* — it is just one of the shapes that achieves a good trade-off between bias and variance; both shapes are presented in Figure 7.

The bias-variance framework reveals that to perform well, a scheduler must minimize both the bias and variance loss contributions. However, we empirically observe a trade-off between the two. We reproduce the bias-variance plot for a smaller model and a different dataset. The results align with the discussion in this section and are presented in App. D.

## 4.2 Data Points Bias

We also explored the possibility of using data points bias as a basis for the bias-variance plot. This idea stems from the notion that different cooldown schedulers can affect bias towards recent data points differently. From this perspective, the best cooldown scheduler would be the one that improves overall performance the most while keeping the bias towards recent data points low. Naturally, a trade-off emerges that is similar to the one we observed before.

To calculate the bias towards data points, we retrospectively calculate the perplexity for previously observed batches in the original order after model training is completed (Bergsma et al., 2025). The results of this evaluation are displayed in Figure 8. Recency bias is evident in both the pre-cooldown model and the post-cooldown models. However, the structure of the bias differs: the pre-cooldown model shows strong recency bias towards recent points, while the post-cooldown models exhibit a U-shaped bias, which is consistent with the findings of Bergsma et al. (2025). This can be explained by the fact that the final cooldown steps are performed with a near-zero learning rate, making the impact of the final data points negligible.

Overall, the structure of the recency bias is greatly controlled by the cooldown shape. We can quantify it using the following framework: Let $p(b_i)$ be the pre-cooldown model perplexity on batch $b_i$, and $c(b_i)$ be the post-cooldown model perplexity on batch $b_i$. If $P$ is the number of training batches before the cooldown stage starts and $N$ is the total number of training batches, we can define the *shift* and *deviation* for a specific cooldown shape as:

$$\mathbf{C} = \{i \in \mathbb{N} : P < i \leq N\} \text{ (cooldown batches indices)},$$

$$\mathbf{difference}_i = c(b_i) - p(b_i),$$

$$\mathbf{shift} = \frac{1}{|\mathbf{C}|} \sum_{i \in \mathbf{C}} \mathbf{difference}_i, \tag{2}$$

$$\mathbf{deviation} = \frac{1}{N} \sum_{i=1}^{N} [\mathbf{difference}_i - \mathbf{shift}]^2 .$$

The idea behind this is that shift represents the improvement of the post-cooldown model over the pre-cooldown model, while deviation represents the consistency of this improvement across different data points. We empirically found that shift calculation is better done only over the cooldown portion of the data, as it leads to an unskewed shift-deviation plot and high correlation with the previously observed results. We do not have a firm justification for this, but we believe it is due to the fact that the pre-cooldown model was not trained on cooldown data at all, and therefore the shift is not skewed by the pre-cooldown model's recency bias.

The results of the evaluation are displayed in Figure 8. The structure of the plot is similar to what we observed before. We present additional visualizations in App. E.

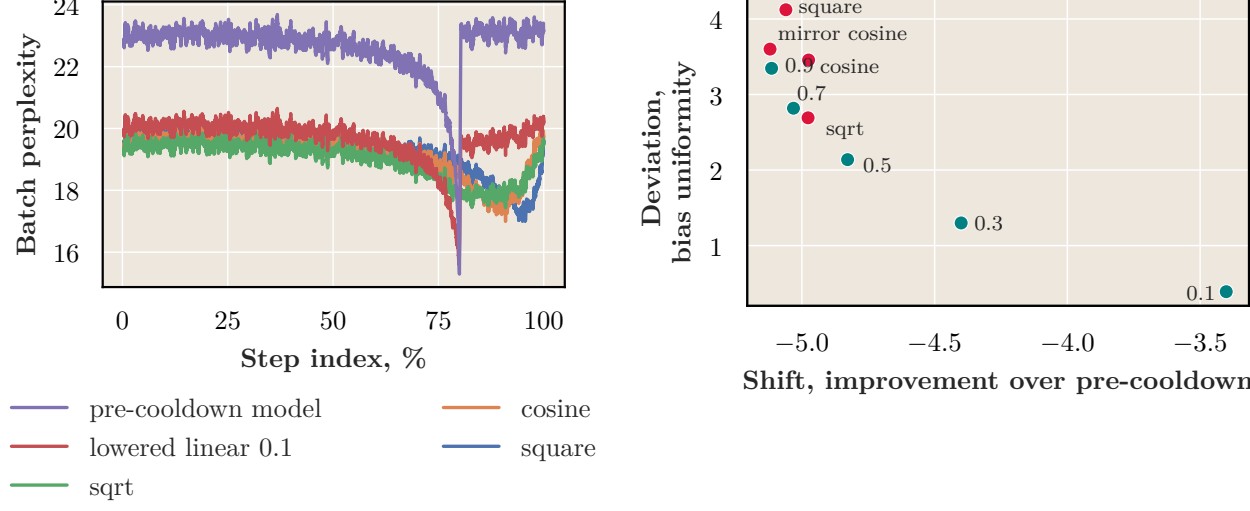

Figure 8: **Shift-deviation result plots. On the left,** evaluation of models on training data after training was completed (100-step window-averaged). It is clear that the pre-cooldown model has pronounced recency bias that ends abruptly at 80% of the total steps (the training limit of the pre-cooldown model). The post-cooldown models have a lower overall mean perplexity and a different recency bias structure. It is clear that the cooldown shape affects the recency bias structure. **On the right,** the shift-deviation plot for different cooldown shapes. It is clear that the pattern is similar to the one we observed before in Figure 6. We explore this similarity further in App. E.

### 4.3 Bias-Variance Plot Conclusions

We conclude that different cooldown shapes regulate a trade-off between bias and variance during cooldown, which significantly impacts performance. Moreover, the same trade-off emerges from the recency bias point of view. Furthermore, we find that there is a set of cooldown shapes, such as *lowered linear 0.7* and *sqrt*, that provide an optimal trade-off. Both cooldown shapes are presented in Figure 7.

## 5 Averaging vs Longer Training

Averaging of models from different runs is a technique extensively used in recent works. In particular, recently released influential models like OLMo2 (OLMo et al., 2025) and Llama3 (Meta AI, 2024) use a technique of performing multiple independent cooldown stages and averaging the final models, a technique also referred to as *model souping* (Wortsman et al., 2022). Another approach is merging completely different runs — a technique explored in Command A model training (Cohere et al., 2025).

With this motivation, we investigate how averaging final models compares to longer training for different cooldown shapes. Specifically, we perform a cooldown stage four times on non-intersecting dataset portions from the same pre-cooldown model. Additionally, we train one model for longer on all data observed by the shorter runs. The longer run uses the *sqrt* cooldown shape for a total of 52.8k steps and a 20% cooldown portion. In total, the longer run achieves the same flops and data portion as four smaller runs combined.

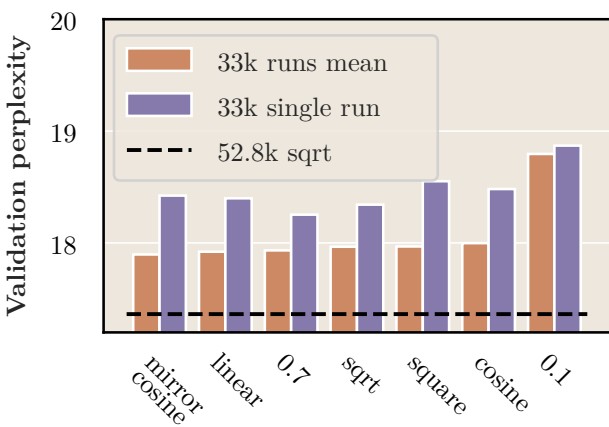

Figure 9: **Performance comparison of averaged models to longer training.** High-variance cooldown shapes yield the best performance. However, a simple longer run outperforms all averages.

Then, we average model weights trained on different dataset portions for each cooldown shape and calculate performance on validation data. The results are displayed in Figure 9. While all averages perform better than their corresponding non-averaged runs, they still fall short of a simple longer run (52.8k, dashed line in Figure 9). It is also evident that high-variance, low-bias shapes provide the best performance when averaging. We consider the possibility that averages perform worse than the longer run due to the model underfitting the data, and longer training may close the gap or even surpass it.

Therefore, in the setting of model averaging, high-variance low-bias shapes like *mirror cosine* perform better than trade-off shapes like *sqrt*. We also discover that a bias-variance plot emerges in model weight space, which we discuss in App. C.

## 6 Cooldown Stage Hyperparameters

While tuning hyperparameters is arguably infeasible for large model runs that are extremely expensive and are often only performed once, the cooldown represents a significantly shorter part of training and can potentially be tuned. Therefore, to better understand the impact of modifying the cooldown shape, we examine the significance of hyperparameter selection during the cooldown stage. This investigation could provide insights into the relative importance of various factors when tuning the model's training. It is divided into two parts: batch size selection and Adam hyperparameter selection.

### 6.1 Batch Size Impact

We use the *sqrt* cooldown shape and vary the batch size, adjusting the number of steps to maintain the same total token count. Additionally, as described by Chiley et al. (2019); Busbridge et al. (2023); Pagliardini et al. (2024), we adjust the AdamW $\beta_1 = 0.9$ and $\beta_2 = 0.95$ parameters to match the optimizer's statistics token half-life. In essence, the token half-life is the number of successive previous steps that contribute to half of the optimizer's EMA accumulator. For a new batch size $\widehat{B} = kB$, the momentum is set as $\widehat{p} = p^k$. We perform simple experiments without adjusting $\beta_1$ and $\beta_2$ (Figure 10b). For experiments with higher batch

sizes, we also increase the learning rate by trying a set of several values and selecting the best one based on performance. The table of learning rate selection for each batch size can be found in App. H; the scaling roughly follows a square root scaling factor (Malladi et al., 2024).

**Batch size variation conclusions.** We observe in Figure 10a that increasing the batch size with matching token half-life leads to deterioration in performance. Therefore, while increasing the batch size can negatively impact performance, we believe this aligns with observations about critical batch sizes described by McCandlish et al. (2018). Despite the unusual loss behavior during the cooldown stage, it appears that batch size variations adhere to patterns previously observed in the literature.

Interestingly, significant performance improvements are observed when no adjustments are made to $\beta_1$ and $\beta_2$ (Figure 10b). This indicates that the token half-life of AdamW states may not be optimal in our setting, a topic further explored in Section 6.2.

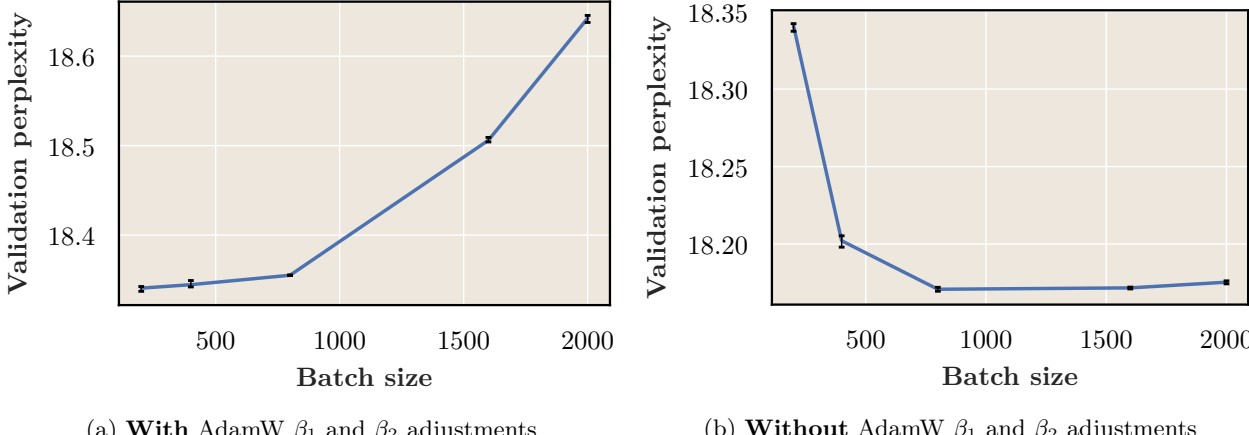

(a) **With** AdamW $\beta_1$ and $\beta_2$ adjustments      (b) **Without** AdamW $\beta_1$ and $\beta_2$ adjustments

Figure 10: **Impact of batch size variation during the cooldown stage on validation perplexity.** Each batch sample contains 512 tokens, and the base batch size is 200. The learning rate is scaled up with increasing batch sizes, which we discuss in App. H. We also plot error bars displaying the minimum and maximum performance of experiments for 5 different data permutations of the same data. **On the left**, besides modifying the batch size, AdamW parameters $\beta_1$ and $\beta_2$ are adjusted to match the token half-life, resulting in performance deterioration for high batch sizes. **On the right**, the batch size is increased without modifying the AdamW parameters, which leads to substantial improvement. This highlights the importance of the AdamW token half-life.

## 6.2 AdamW Parameters Impact

We explore how AdamW betas impact cooldown stage performance and compare the resulting performance improvements to those achieved through shape selection. For all experiments, we use the *sqrt* cooldown shape and vary the tokens' half-life by adjusting $\beta_1 = 0.9$ and $\beta_2 = 0.95$, modifying both using the parameter $p$: $\widehat{\beta_1} = \beta_1^p, \widehat{\beta_2} = \beta_2^p$. The results are presented in Figure 11. In addition to modifying both $\beta_1$ and $\beta_2$, we conduct experiments with $\beta_2$ variation while keeping $\beta_1 = 0.9$ fixed. These results are also shown in Figure 11.

**Tokens Half-Life Variation Conclusions.** We observe that varying the AdamW tokens' half-life can lead to significantly different performance outcomes. From the experiment with varying both $\beta_1 = 0.9$ and $\beta_2 = 0.95$, we see that the best cooldown shape at $p = 1$ is almost identical, in terms of perplexity, to the worst at $p = 0.3$. Therefore, modifying AdamW parameters during cooldown is worthwhile, as performance improvements can be comparable to those achieved through cooldown shape selection.

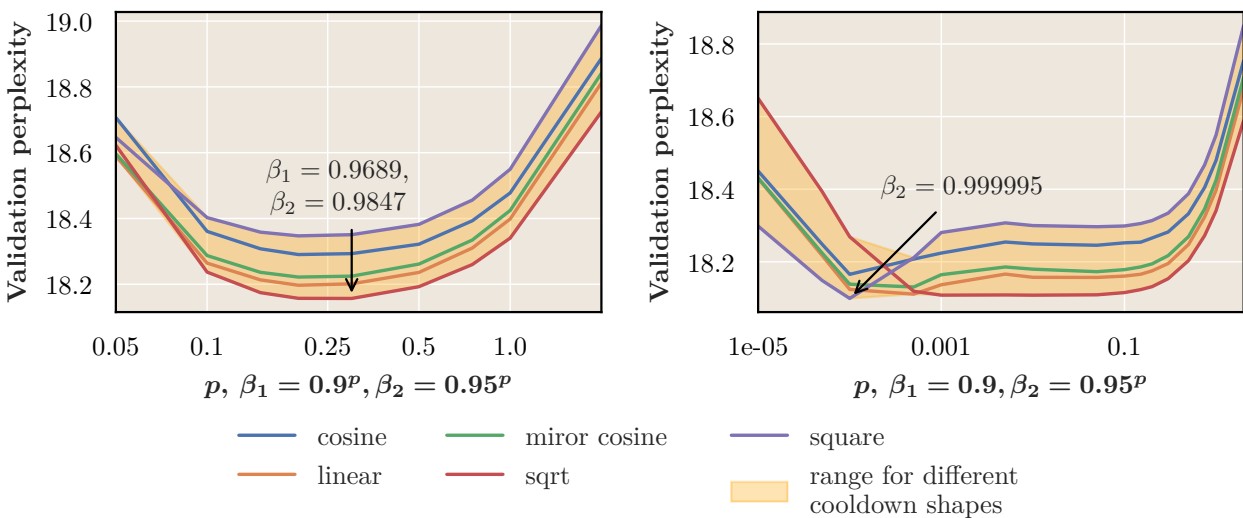

Figure 11: **Evaluation of the impact of AdamW parameters on cooldown performance. On the left,** we vary both $\beta_1$ and $\beta_2$. **On the right,** we vary only $\beta_2$. The betas are adjusted using the parameter $p$: $\widehat{\beta} = \beta^p$, with base betas $\beta_1 = 0.9$ and $\beta_2 = 0.95$. Examining the range for different cooldown shapes makes it clear that a good choice of hyperparameters can greatly improve performance, matching or even surpassing the importance of cooldown shape selection.

From the second plot in Figure 11, it is noticeable that performance greatly depends on $\beta_2$ variations, with surprisingly large values tending to yield the best results.

Additionally, a notable property is that the optimal cooldown shape, from a bias-variance perspective, remains unchanged with the choice of $\beta_1$ and $\beta_2$ within a reasonable range of values. This is clearly seen in Figure 11, as *sqrt* acts as a lower bound for all the shapes tested unless $p$ becomes too small. In the case of small $p$, the AdamW tokens' half-life becomes so large that the optimizer state is almost not updated, which is not a reasonable setting and requires higher learning rates.

## 7 Examining the Loss Landscape

The "river valley perspective" is one of the intuitions behind the WSD scheduler proposed by Wen et al. (2024), suggesting that WSD optimization can be visualized as descending from a mountain top along a river. The constant learning rate stage corresponds to descending in the general direction of the river's flow, while the cooldown stage corresponds to descending directly into the river itself. However, a clear visualization of the river valley is still lacking. This section focuses on visualizing the river valley during the WSD cooldown stage.

To achieve this, we plot the loss landscape using the following coordinates:

$e_1$ A vector in weight space connecting the pre-cooldown checkpoint to the final model after cooldown. We refer to this as the *global optimization* direction.

$e_2$ A vector corresponding to ten optimizer steps from a given point. We refer to this as the *Adam steps* direction. We choose ten steps to minimize the impact of noise on the obtained vector.

**Loss Landscape Plots Interpretation.** Figures 12 and 30 present the results. These plots clearly show a river valley along the global optimization direction at the beginning of the cooldown stage. It remains visible in the middle of the cooldown stage, though less pronounced as optimization nears descent into the

basin. By the end of the cooldown stage, only the final basin is visible. We believe these plots provide strong evidence supporting the river valley basin concept.

Moreover, this perspective provides further insights for the bias-variance framework as the observation that a local optimization step (from a single example) is almost orthogonal to the best global direction is exactly what causes the variance. Training cannot make progress downstream (reducing the bias) without taking large steps along the orthogonal directions (causing variance).

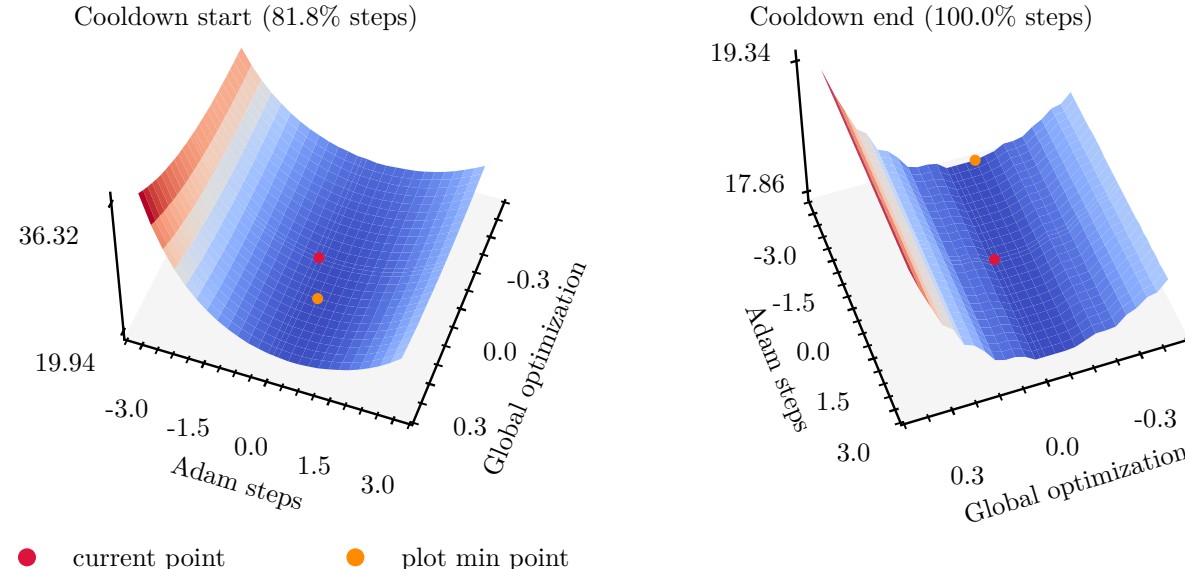

Figure 12: **Loss landscape plots at different cooldown points.** We plot the loss landscape using two vectors: the global optimization direction (the line between the pre-cooldown checkpoint and the final model) and the Adam steps direction (the line between the given point and ten optimizer steps from it). The Z-axis represents perplexity on a portion of validation data. We see a potential valley along the global optimization direction, which is less pronounced as optimization nears descent into the basin. A larger plot is presented in Figure 30.

## 8 Related Work

**Learning rate schedulers.** The evolution of learning rate schedulers has been driven by the need to adjust the learning rate during training to improve optimization convergence. This need can also be viewed as a matter of balancing exploration and exploitation (Iyer et al., 2020; Subramanian et al., 2024). Traditional approaches, such as step decay (reducing the learning rate at fixed intervals) and exponential decay (continuously reducing the learning rate), laid the foundation for schedule-based optimization (Li & Arora, 2019; Ge et al., 2019; Smith et al., 2017). The cosine schedule (Loshchilov & Hutter, 2017) is one of the most common choices in large-scale deep learning. Others include inverse square root (Raffel et al., 2020; Chowdhery et al., 2023), Noam (Vaswani et al., 2017) or stepwise schedules (Bi et al., 2024). In the last year, Warmup-Stable-Decay (WSD), also called the trapezoidal schedule (Dosovitskiy et al., 2020) , has become popular for its benefits of an undetermined training length, performance (Zhai et al., 2022; Shen et al., 2024a; Hägele et al., 2024; Hu et al., 2024) and continual learning (Ibrahim et al., 2024; Gupta et al., 2023; Janson et al., 2025).

Given the significant impact of the learning rate scheduler on model performance, and considering the complexity of finding the optimal learning rate for LLMs due to the interplay between learning rate, batch size,

number of training tokens, model size, and other hyperparameters, multiple works have focused on schedule-free optimization (Defazio et al., 2024) or scheduling techniques that work well with different model sizes (Shen et al., 2024b).

**Understanding dynamics of learning rate schedulers.** The interaction between learning schedules and optimization dynamics remains an active research area. For example, for WSD, Wen et al. (2024) proposed the "river valley" hypothesis to explain WSD behavior, while Subramanian et al. (2024) analyzed phase transitions in loss landscapes during schedule changes. Our work extends these observations through direct visualization of the cooldown-stage loss landscape. Additionally, recent work by Schaipp et al. (2025) showed that the empirical behavior of the WSD scheduler behaves surprisingly similarly to a performance bound from non-smooth convex optimization theory (Defazio et al., 2023) which stems from either non-smoothness or non-vanishing gradient norms. We investigate such gradient norms in the cooldown stage in App. G.

## 9 Conclusions

This work provides a comprehensive analysis of the cooldown stage in the WSD learning rate scheduler, offering insights into its dynamics and impact on model performance. The key findings from this study are summarized below.

**Cooldown Shape Selection.** The shape of the cooldown phase critically influences model performance. We introduce a bias-variance framework to compare different cooldown shapes, identifying *sqrt* and *lowered linear 0.7* as optimal choices due to their ability to balance exploration and exploitation. We observe that high-variance low-bias shapes like *mirror cosine* perform better when final model averaging is performed. We also provide a recency bias perspective that aligns with the bias-variance framework.

**Hyperparameter Sensitivity.** Our experiments show that varying the batch size during the cooldown stage has a relatively minor impact on performance, consistent with prior findings on critical batch sizes. However, adjusting the AdamW hyperparameters ($\beta_1$ and $\beta_2$) revealed significant performance differences, comparable to those observed with cooldown shape selection. Notably, larger values of $\beta_2$ consistently yielded improved results, highlighting the importance of fine-tuning these parameters.

**Loss Landscape Visualization.** We further validate the "river valley perspective" through loss landscape visualizations. These plots illustrate how optimization transitions from broad exploration during the constant learning rate phase to focused descent into a loss basin during the cooldown phase, providing evidence for this conceptual framework.

## 10 Future Work

While our analysis is limited to moderate model sizes, we validate the main findings across different numbers of parameters and datasets (D). For future directions, we believe there remain several important opportunities to deepen our understanding:

1. Our work focuses solely on the validation loss and the optimization landscape. However, future work should explore the relevance of observed effects for downstream tasks and model behavior, although a connection between loss and downstream performance was previously observed (Du et al.; Hägele et al., 2024).

2. Related to the above, it remains unclear what mechanistic changes occur in the model over the course of the cooldown stage. As a first step, we explore internal feature quality improvement over the course of the cooldown stage with experiments in App. I, but we did not arrive at useful conclusions. Other possible approaches include the recently proposed model diffing frameworks (Bricken et al., 2024; Lindsey et al., 2024).

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

# A  Experiments Context

The experimental setup mirrors that of Hägele et al. (2024). Specifically, we employ a decoder-only transformer model resembling Llama3 (Meta AI, 2024). The architecture incorporates SwiGLU activations (Shazeer, 2020), RoPE (Su et al., 2023), RMSNorm (Zhang & Sennrich, 2019), and alternating attention and feed-forward layers. Unless explicitly stated otherwise, we adhere to the following methods and hyperparameters.

We use the AdamW optimizer with beta parameters set to $(\beta_1, \beta_2) = (0.9, 0.95)$, $\varepsilon = 10^{-8}$, a decoupled weight decay of 0.1 (Loshchilov & Hutter, 2019), and gradient clipping at 1.0. All runs are trained with *bfloat16* automatic mixed precision. We generally apply a short warmup of 300 steps. The batch size is set to 200, corresponding to 0.1 million tokens for a sequence length of 512. The vocabulary is derived from the GPT-2 tokenizer (Radford et al., 2019) and consists of 50,304 tokens. Hidden shapes of the used model are presented in Table 1.

| Model Size | $d_{\mathbf{model}}$ | $n_{\mathbf{layers}}$ | ffw dim | head dim | $n_{\mathbf{heads}}$ |
|:---:|:---:|:---:|:---:|:---:|:---:|
| 60M | 512 | 10 | 1536 | 64 | 8 |
| 210M | 768 | 24 | 2048 | 64 | 12 |

Table 1: Model configurations with varying parameter counts. The 210M parameter model is used in the majority of the experiments. The 60M parameter model is used for experiments in App. D.

# B  Used Cooldown Shapes Formulas

The formulas for cooldown shapes used in the article and presented in Figure 4 are:

$$x \in [0, 1],$$
$$\mathbf{linear} = 1 - x,$$
$$\mathbf{cosine} = \frac{1 + \cos(\pi x)}{2},$$
$$\mathbf{mirror\ cosine} = 2(1 - x) - \frac{1 + \cos(\pi x)}{2},$$
$$\mathbf{square} = 1 - x^2,$$
$$\mathbf{sqrt} = 1 - \sqrt{x}.$$

# C  Bias-Variance Plot Basis Selection

In addition to the basis for bias-variance plots proposed in equation (1) (**loss-based space**), we analyze how results change with the selection of different bases.

## C.1  Model Weights Space

Surprisingly, the same structure is observed when results are plotted in model weights space. We use each experiment's model weights as a vector of model parameters of size $p$. That is, if $s_{in} \in \mathbb{R}^p$ represents model weights trained with cooldown shape $i$ by data permutation $n$, and $r_n \in \mathbb{R}^p$ represents reference model weights trained on data permutation $n$, the complete formulas used are presented in equation (3). In essence, we measure bias as the distance between the mean reference and experimental models, and variance as the variance of model weights across different permutations.

$$\mathbf{bias}_i = \left\| \frac{1}{N} \left( \sum_{n=1}^{N} r_n - \sum_{n=1}^{N} s_{in} \right) \right\|,$$

$$\mathbf{variance}_i = \frac{1}{N-1} \sum_{n=1}^{N} \|s_{in} - \overline{s_i}\|^2.$$

(3)

Plots in this basis are presented in Figure 13. The experiments used are the same as in section 4.1, but with different coordinates. We also plot a comparison of the two spaces in Figure 14. We see that while some variations are observed, the general structure of the plot remains the same. Presumably, variations between plots should become less noticeable with an increase in the number of experiments with different data permutations.

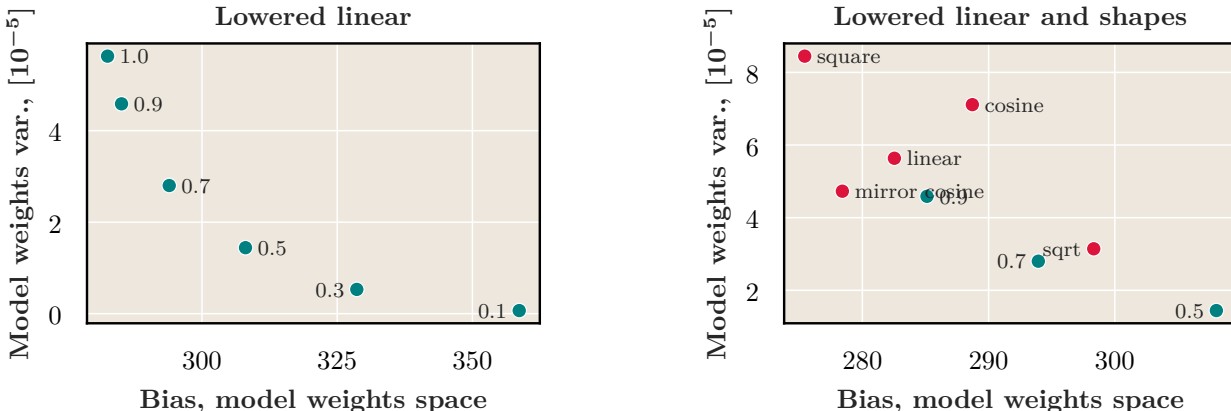

Figure 13: **Bias-variance plot in model weights space.** The general structure agrees with the discussion in section 4.1.

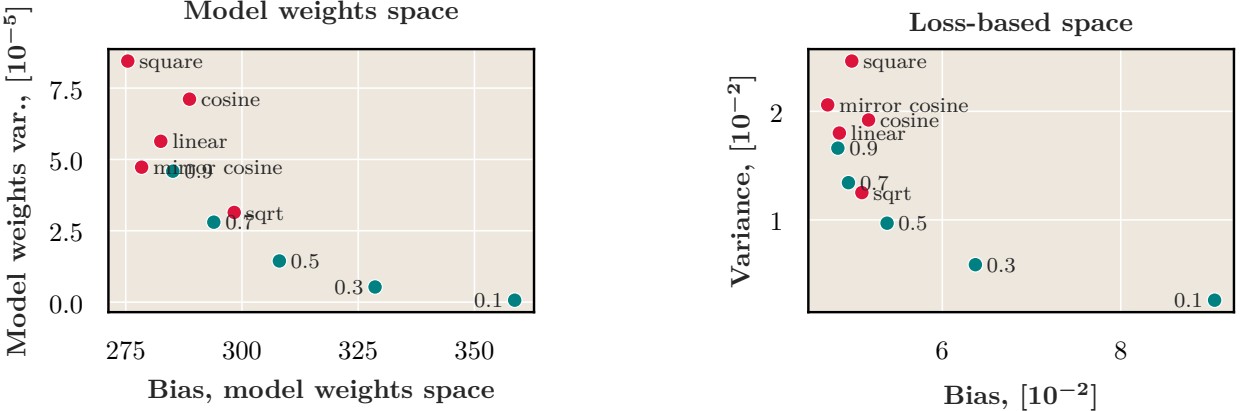

Figure 14: **Comparison of bias-variance plots in loss space and model weights space.** While minor variations are noticeable, the general structure of the plots is the same.

## C.2 Other Loss-based Spaces

Another coordinate space used for the bias-variance plot is a simple loss-based space. If $s_{in} \in \mathbb{R}$ represents the model validation loss trained with cooldown shape $i$ by data permutation $n$, the complete formulas used are presented in equation (4), which are simply the mean and variance.

$$
\begin{aligned}
\mathbf{bias}_i &= \frac{1}{N} \sum_{n=1}^{N} s_{in}, \\
\mathbf{variance}_i &= \frac{1}{N} \sum_{n=1}^{N} \|s_{in} - \overline{s_i}\|^2.
\end{aligned}
\tag{4}
$$

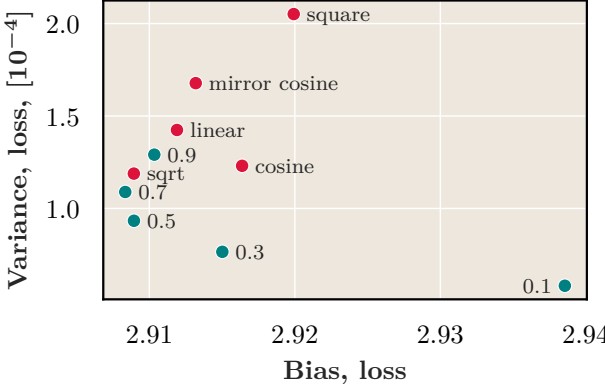

Figure 15: Comparison of the bias-variance plot in loss coordinates. While the general idea of the bias-variance trade-off is noticeable, the plot appears skewed.

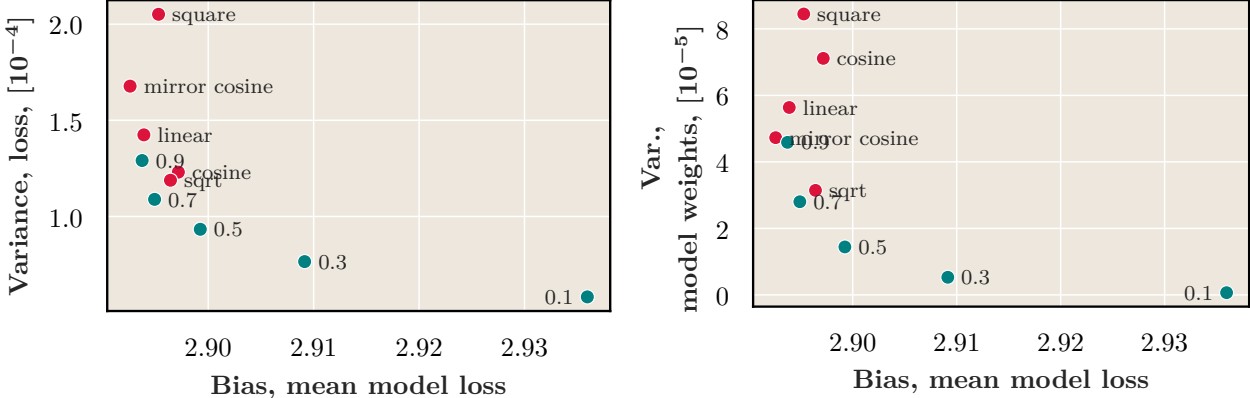

(a) Bias-variance plot where **bias** represents the **loss of the average model** and **variance** is the **variance of losses** for each specific shape and different data permutations.

(b) Bias-variance plot where **bias** represents the **loss of the average model** and **variance** is the **variance of model weights** for each specific shape and different data permutations.

Figure 16: **Miscellaneous loss-based bias-variance plots.**

Plots in this basis are displayed in Figure 15. The observed trade-off is not as vivid as in loss-based or model weights coordinates and appears to be a skewed version of the plots observed before.

We also tried using the loss of the average model as bias instead of the mean of the losses, and the variance of the losses (Figure 16a). Additionally, we tried using the loss of the average model as bias and the variance of model weights as variance for each experiment (Figure 16b). While both of these plots produce reasonable results, we believe they are less interpretable than the approach used in the main part of the paper.

### C.3 KL-based Space

We evaluate models with different cooldown shapes and data orderings on the validation dataset portion. For each of these models, we calculate the $KL$ divergence to the average prediction of $N$ reference models (referred to as single *reference model* predictions) and determine the variance of the predictions from the $N$ models. That is, for vocabulary size $V$, total tokens count $K$, if $s_{ink} \in \mathbb{R}^V$ represents model predictions (soft-labels) trained with cooldown shape $i$ by data permutation $n$ for token $k$, and $r_{nk} \in \mathbb{R}^V$ represents reference model predictions (soft-labels) trained on data permutation $n$ for token $k$, the complete formulas used are presented in equation (5).

$$
\begin{aligned}
\mathbf{bias}_i &= \frac{1}{K} \sum_{k=1}^{K} KL\left( \frac{1}{N} \sum_{n=1}^{N} r_{nk}, \frac{1}{N} \sum_{n=1}^{N} s_{ink} \right), \\
\mathbf{variance}_i &= \frac{1}{NK} \sum_{n=1}^{N} \sum_{k=1}^{K} \| s_{ink} - \overline{s_{ik}} \|^2.
\end{aligned}
\tag{5}
$$

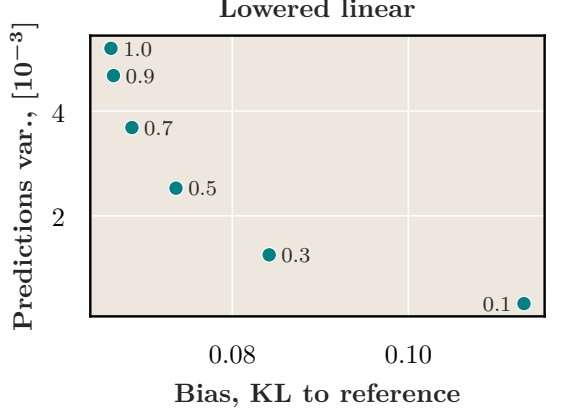
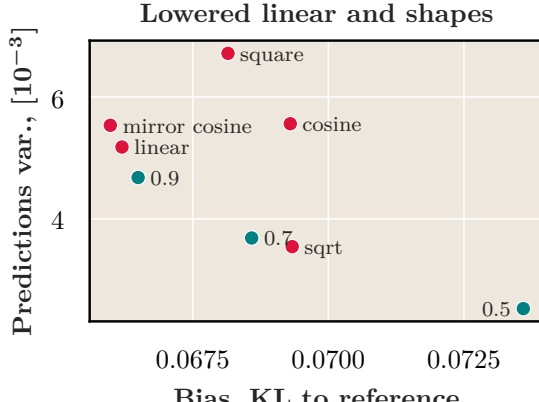

Figure 17: **Bias-variance plot for different cooldown shapes.** We use the KL divergence to measure *bias* as the average distance between predictions on the validation dataset portion of a reference model (average predictions of models trained for longer on the same data) and an experimental model's mean predictions. *Variance* is the variation in predictions for experiments with different data orderings (Equation 5). **On the left,** we compare only lowered linear shapes (Figure 5), which exhibit an expected decrease in variance and increase in bias with decreasing parameter value. **On the right,** the full range of nonlinear shapes along with selected lowered linear shapes for reference. The *lowered linear* shape with parameter 0.7 and the *sqrt* shape occupy favorable positions, achieving a balanced trade-off between bias and variance.

The experimental results are presented in Figure 17. While the result is similar to the basis space used in the main text, we believe that coordinates choice is less interpretable and therefore we are not using it as the main version.

# D    Bias Variance Plot Reproduction

We reproduced the bias-variance plot for a smaller model with 60M parameters and for a different dataset. The observed results align with previous experiments presented in the paper.

## D.1    Smaller Model

We reproduced the bias-variance plot for a smaller model with 60M parameters. We used the same architecture as in the main experiments but modified the number of layers and the hidden dimension. The results are presented in Figure 18 and Figure 19. From the plots, it is evident that the general structure and relative positions of different shapes agree with the previous experiments.

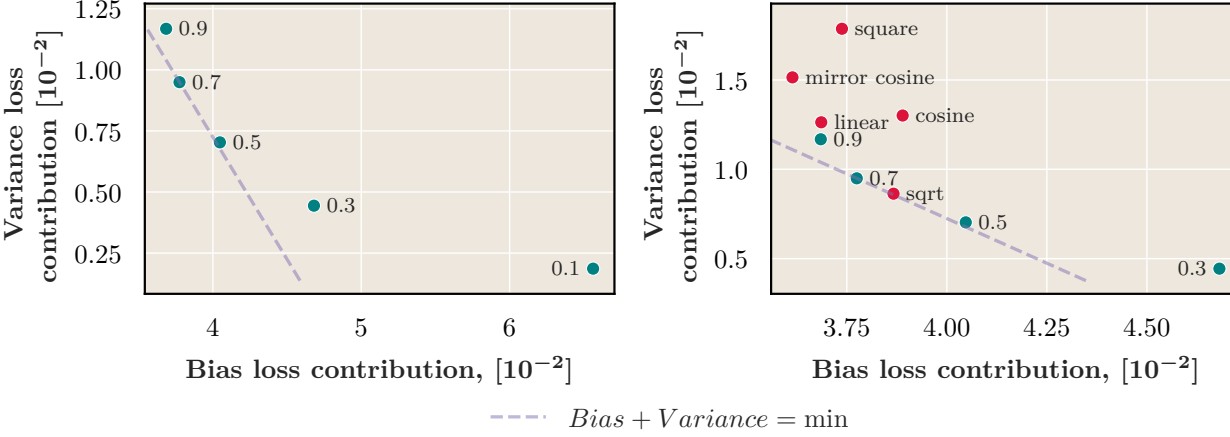

Figure 18: Bias-variance plot for different cooldown shapes for the smaller model **in loss-based space**. The observed results agree with our earlier discussion in the paper.

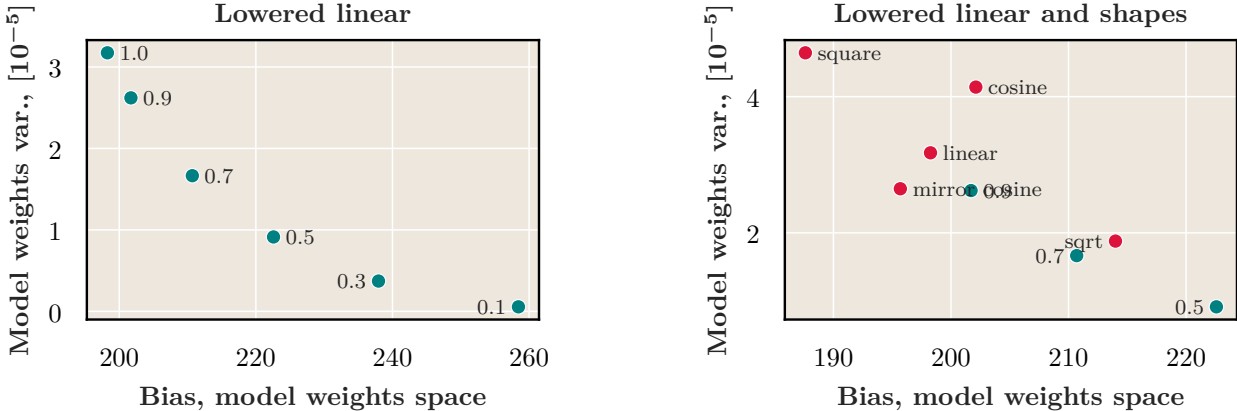

Figure 19: Bias-variance plot for different cooldown shapes for the smaller model **in model weights space**. The observed results agree with our earlier discussion in the paper.

## D.2 Different Dataset

Additionally, we reproduced the bias-variance plot using a different dataset and the same model. We used the fineweb-edu (Lozhkov et al., 2024) 10 billion token subset. The results are presented in Figure 20 and Figure 21.

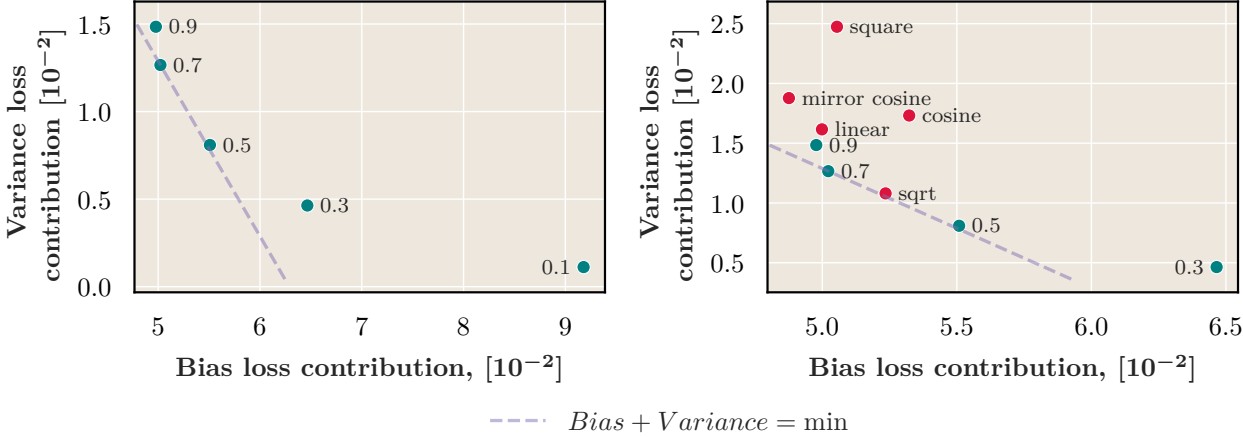

Figure 20: Bias-variance plot for different cooldown shapes for the fineweb-edu dataset **in loss-based space**. The observed results agree with our earlier discussion in the paper.

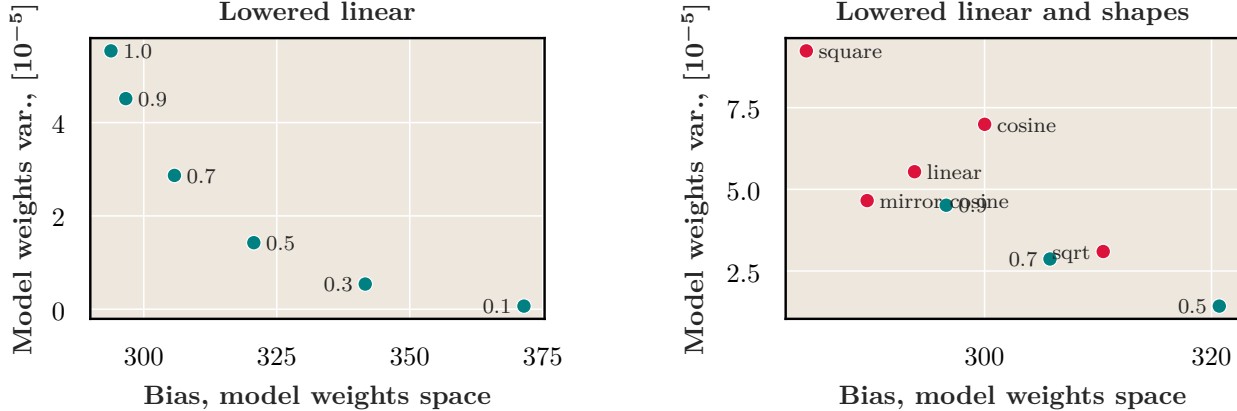

Figure 21: Bias-variance plot for different cooldown shapes for the fineweb-edu dataset **in model weights space**. The observed results agree with our earlier discussion in the paper.

## E    Data Points Bias Supplements

We present additional visualizations for the recency bias-based shift-deviation framework in this section. Figure 22 shows the unreduced components from equation 2.

We also explore how well the shift-deviation plot (Figure 8) aligns with the bias-variance plot (Figure 6). The correlation is presented in Figure 23. It is clear that both perspectives align well. The plot in shift-deviation coordinates, with shift calculated over the entire training process, is presented in Figure 24. The correlation with bias is not as strong, but the general structure remains the same, though skewed.

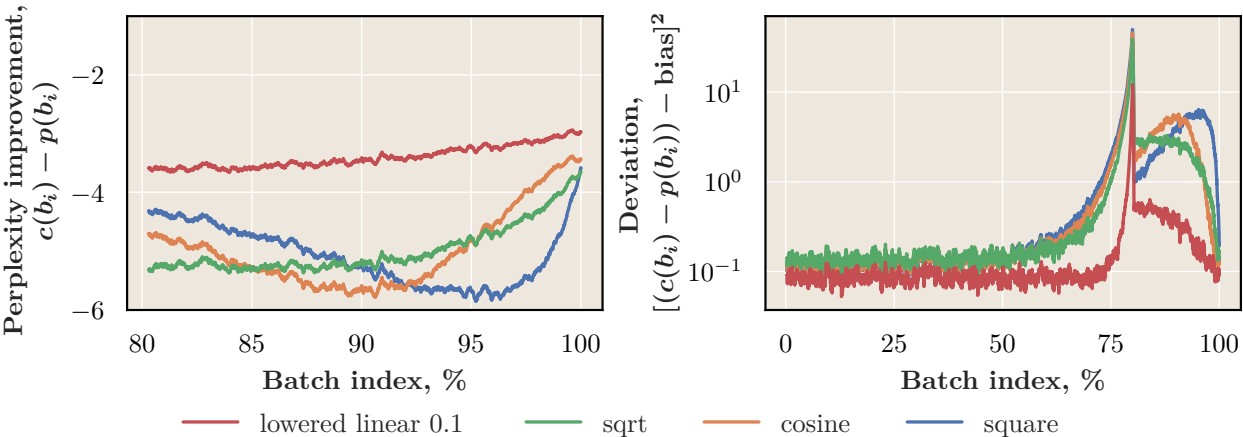

Figure 22: **Bias-variance statistics from equation 2. On the left,** unreduced and 100-batch window-smoothed shift from eq. 2 (calculated only during cooldown). **On the right,** the unreduced and 100-batch window-smoothed deviation from eq. 2. It is clear that low-deviation shapes such as *lowered linear 0.1* lie close to zero on the right plot, indicating more uniform data point bias. However, on the left plot, they lie high, indicating low improvement over the pre-cooldown model (shift). Conversely, high-deviation shapes such as *square* have low shift and high deviation.

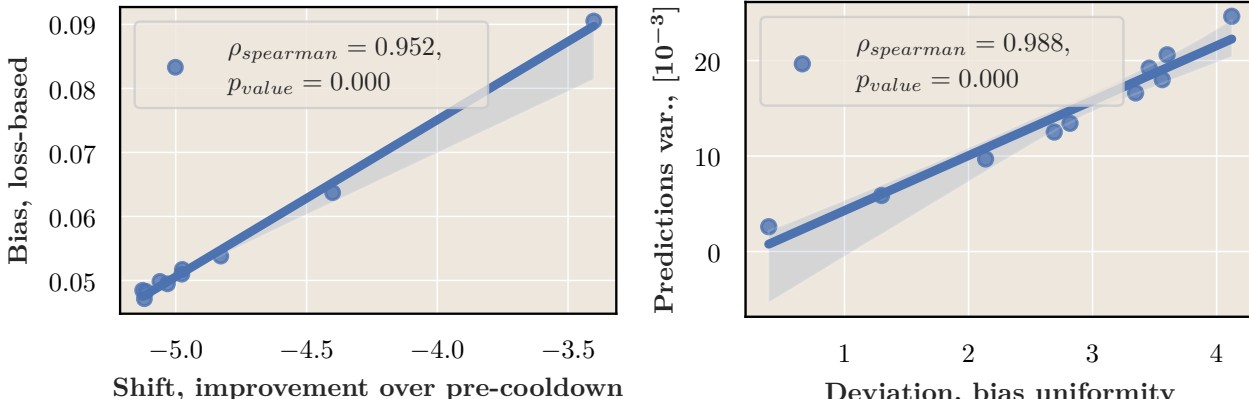

Figure 23: **Comparison between bias-variance and shift-deviation plots.** The left plot shows the correlation between shift and bias, while the right plot shows the correlation between deviation and variance. The correlation is calculated using the Spearman rank correlation coefficient. High correlation is clearly visible, indicating that the two perspectives align well.

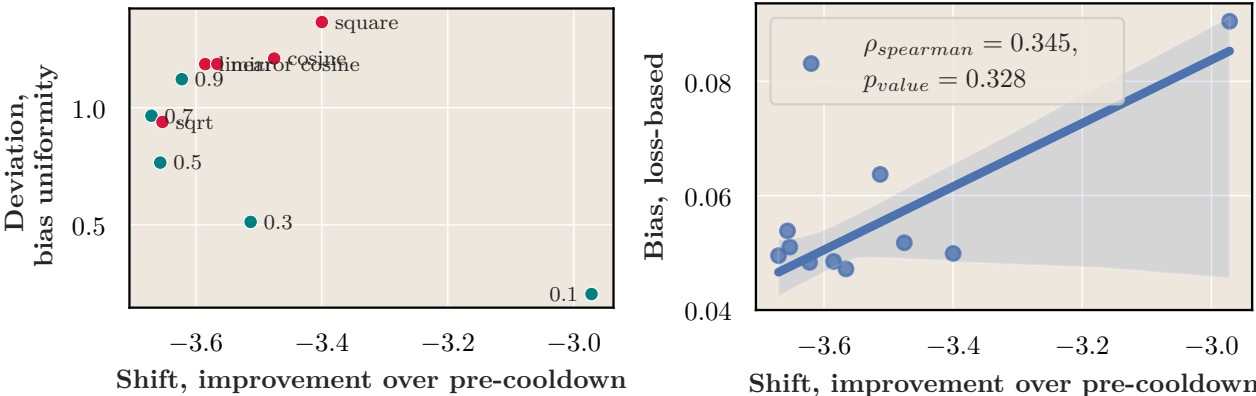

Figure 24: **Shift-deviation plot with shift calculated over the entire training process. On the left,** the shift is calculated over the entire training process, and **on the right,** the resulting correlation with bias is presented. It is noticeable that the correlation with previous results is not as strong, but the general structure remains the same. Also, unwanted skew is visible on the left plot.

## F  Weight Decay and Optimizer State

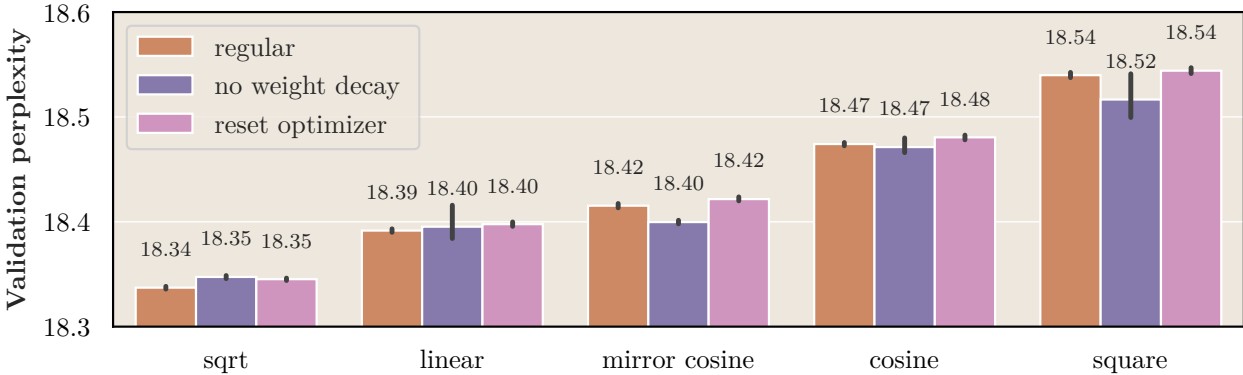

Figure 25: **Comparison of cooldown performance without weight decay or with an optimizer state reset to usual runs.** For the *sqrt* shape, disabling weight decay or resetting the optimizer slightly worsens performance, while for high-variance shapes, disabling weight decay can improve performance.

In all the experiments, we use AdamW with a weight decay of 0.1 and resume training from the pre-cooldown stage, restoring the optimizer state. It is worth examining how the cooldown stage performs without weight decay or with a reset optimizer state. Figure 25 presents the results of these experiments, grouped by different cooldown shapes.

**No Weight Decay and Reset Optimizer State Performance Conclusions.** Switching off weight decay or resetting the optimizer state negatively impacts bias-variance optimal shapes like *sqrt*, although the effect is not dramatic, as evident from the absolute values of the difference.

Interestingly, switching off weight decay improves performance for high-variance shapes such as *square* or *mirror cosine*. This can be explained by introducing an "effective learning rate", as discussed by Kosson et al. (2024a). For high-variance shapes, switching off weight decay acts as an additional decaying factor for the learning rate, altering the shape's position on the bias-variance plot.

## G    Momentum & Gradient Alignment

Motivated by the idea that the cooldown stage optimization exhibits some unusual properties, we explore how gradients align with the AdamW momentum state during the cooldown stage. To do this, we calculate the cosine angle between the optimizer's state and the corresponding gradient during training for each parameter individually (Figure 26a). Additionally, we collect gradient norms during the cooldown stage (Figure 26b).

**Gradient Alignment Conclusions.**   We observe that alignment with momentum increases during the cooldown stage, indicating that gradients become more aligned between optimization steps: starting from negative alignment with momentum and moving to slightly positive. This rising alignment suggests that the optimizer's state and the gradients converge toward a more stable direction as training progresses through the cooldown phase (Becker et al., 2024; Liang et al., 2025).

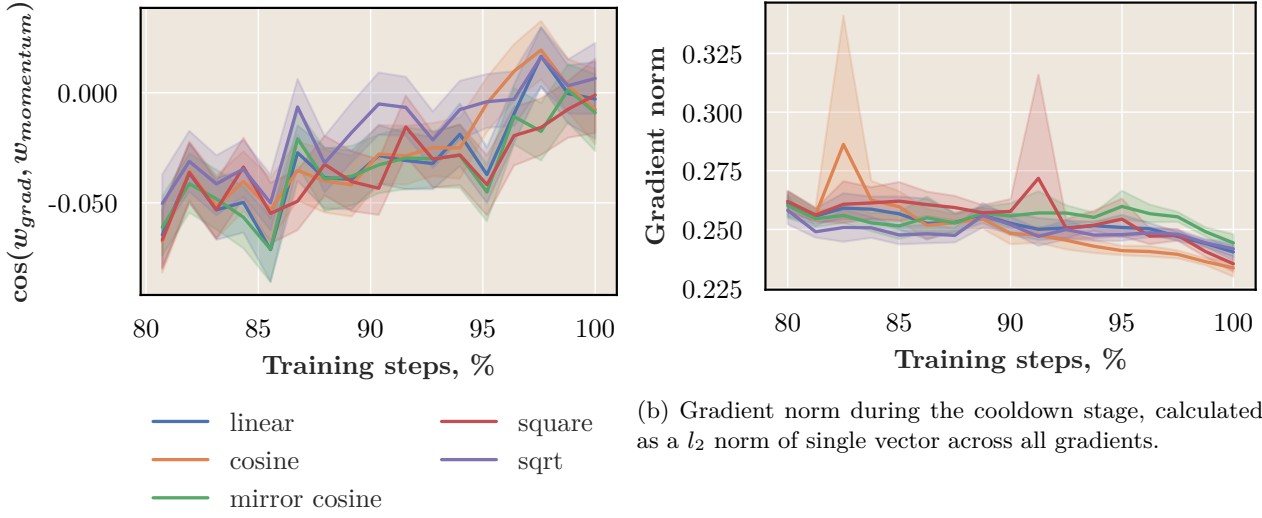

(b) Gradient norm during the cooldown stage, calculated as a $l_2$ norm of single vector across all gradients.

(a) Average cosine similarity between gradients and the corresponding AdamW momentum state (*exp_avg*).

Figure 26: **Gradient statistics during the cooldown stage.** Confidence bands represent the 95% confidence interval. Values are averaged using a sliding window of 400 steps.

## H    Batch Sizes Sweep Learning Rate Selection

We tried several higher learning rates for larger batch sizes in experiments of Section 6.1 and selected the best one based on final performance. The Table 27 presents learning rate scaling factors (based on pre-cooldown stage learning rate) that were chosen for the presented experiments. The scaling roughly follows a square root scaling factor (for a batch size $k$ times larger, a scaling of $\sqrt{k}$ is chosen) (Malladi et al., 2024).

| Batch Size | Learning Rate Scaling Factor |
|---|---|
| 200 | 1.06 |
| 400 | 1.50 |
| 800 | 2.12 |
| 1600 | 3.00 |
| 2000 | 3.35 |

Figure 27: Learning rates scaling factor (based on pre-cooldown stage learning rate) for different batch sizes.

# I    Features Evolution Experiments

We investigate how output features from different transformer layers evolve during training. To analyze this, we apply linear probes (Razzhigaev et al., 2024; Tomihari & Sato, 2024) during the cooldown stage at intervals of 1000 steps across all transformer hidden layers. The linear layers are trained for 2,000 steps (1 million tokens) using the AdamW optimizer on a subset of the training data, then evaluated on 1,000 evaluation batches from a held-out portion of the training dataset. Linear probing weights are initialized with the current lm-head weight.

**Results discussion.** Figure 28 demonstrates progressive improvements in linear probe perplexities throughout the cooldown phase across layers. Figure 29 reveals the temporal dynamics of these changes, showing that the final layers exhibit the most significant relative improvement during cooldown. This pattern suggests that deeper layers undergo more substantial feature refinement in later training stages compared to earlier layers.

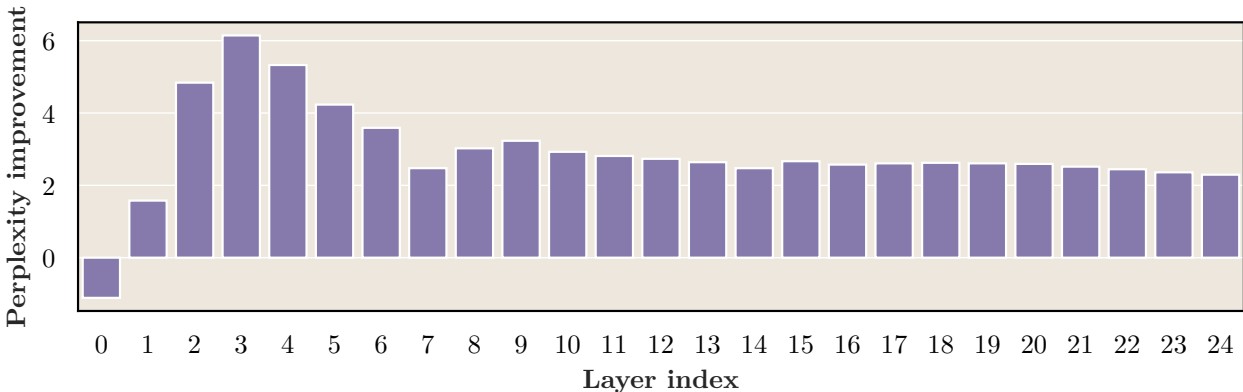

Figure 28: **Perplexity improvement over the course of the cooldown stage for linear probing of different hidden transformer layers.** The zero layer corresponds to embeddings.

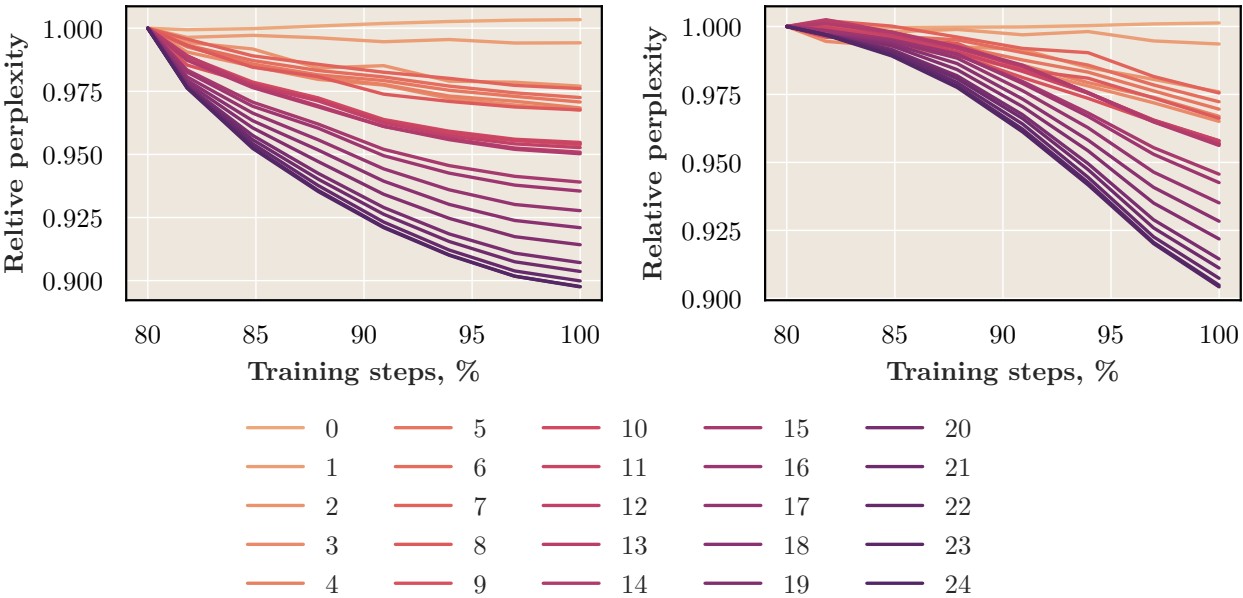

Figure 29: **Dynamics of perplexity improvement over the course of the cooldown stage for linear probing of different hidden transformer layers.** The zero layer corresponds to embeddings. Perplexity relative to the start of the cooldown stage is reported. **Left** uses a *sqrt* cooldown shape, **right** uses a *square* cooldown shape.

## J    Loss Landscape Plots

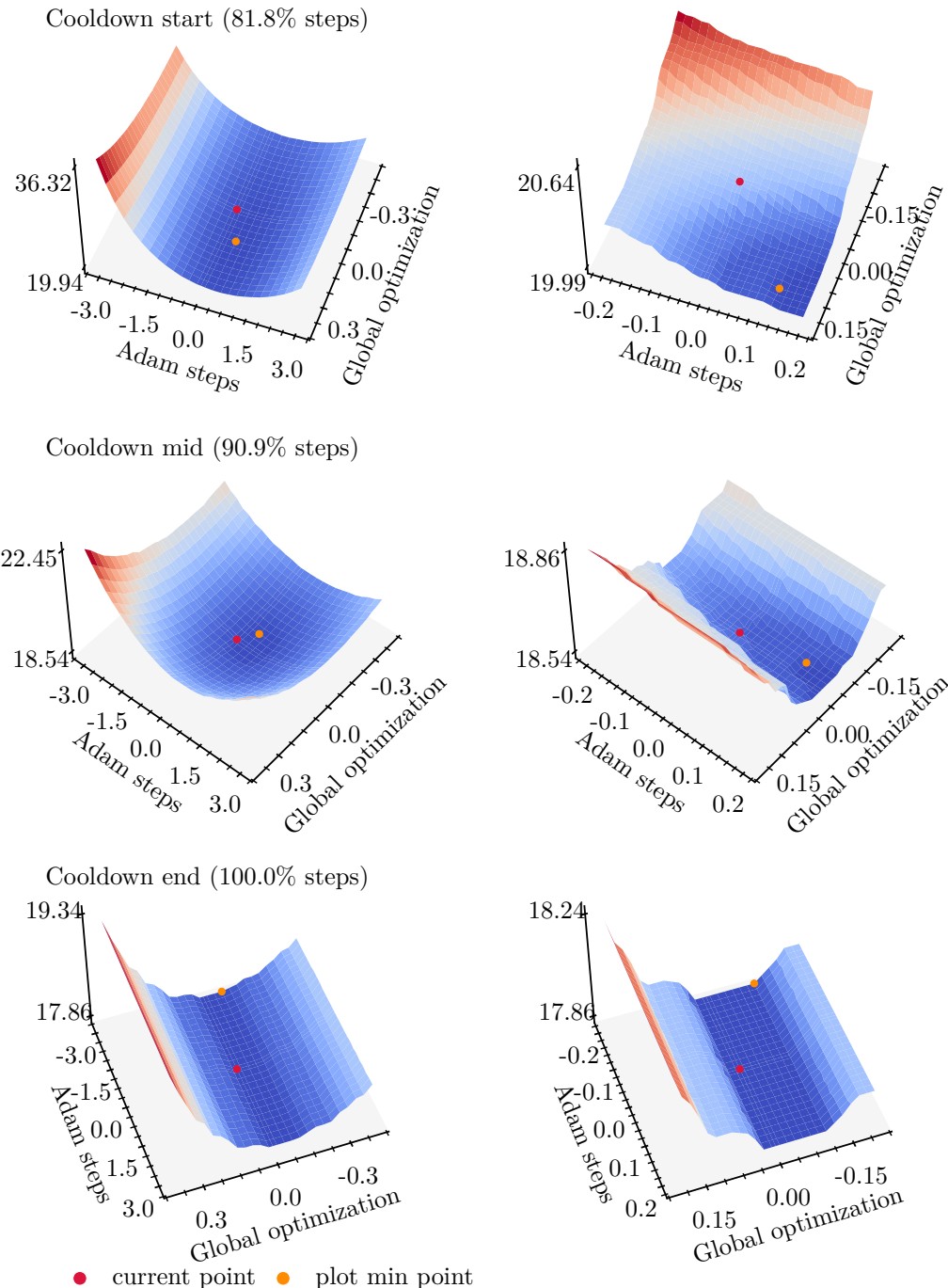

Figure 30: **Plots of the loss landscape at different cooldown points.** The Z-axis represents perplexity on a portion of the validation data. On the left, plots from a larger scale are displayed; on the right, the same plot points are shown at a smaller scale.

