# OpenReview forum: "Training Dynamics of the Cooldown Stage in Warmup-Stable-Decay Learning Rate Scheduler"
_TMLR — Accepted by TMLR_

### Review · Reviewer_v6TE · 2025-05-05

**Summary Of Contributions:**

This paper investigates the role of the cooldown stage in the Warmup-Stable-Decay (WSD) learning rate scheduler, a phase often overlooked in prior work. The authors introduce a bias-variance framework to analyze how different cooldown shapes impact model performance, finding that shapes like sqrt and lowered linear 0.7 offer strong trade-offs. They complement this with experiments on optimizer hyperparameters and visualize the loss landscape to support the "river valley" hypothesis. The study provides practical guidance for tuning the cooldown phase to improve transformer training outcomes.

**Audience:**

Yes

**Broader Impact Concerns:**

No ethical concerns.

**Claims And Evidence:**

Yes

**Requested Changes:**

1. The discussion on model averaging versus longer training stops short of offering a broader conclusion. It would be helpful to include a take-home message about when averaging is or isn't worthwhile.
2. The notion of recency bias is intriguing, but currently, the conclusions feel tentative. Consider including a deeper analysis or a concrete hypothesis to explain the emergence of the U-shaped perplexity profile during cooldown. This would sharpen the section’s contribution.
3. Some of the figures, particularly in Section 4, are rich but a bit dense. For example, Figure 9 would benefit from more precise legends.

**Strengths And Weaknesses:**

**Strength**
1. This paper studies an important problem and provides novel insights. The scope is tightly focused. By looking into the cooldown phase in isolation, the paper offers actionable insights for practitioners, especially those tuning large models under fixed compute.

2. The bias-variance framework is intuitive and well-motivated. The authors go beyond simple performance comparisons to propose a conceptual and empirical tool that aids in understanding why certain cooldown shapes are preferable.

**Weakness**
1. The recency bias section, while conceptually interesting, feels underdeveloped. The explanation of why recency bias transitions from sharp to U-shaped is somewhat speculative
2. It is unclear whether the difference in validation perplexity between different cooldown shapes is significant enough to study the problem or not. If the difference is too negligible, people would just choose the random one to use.
3. Would it be possible to include downstream task performance in the evaluation?

---

> ### Author Response · Authors · 2025-06-12
> **Implemented Changes**
>
> Thank you for the feedback!
>
> In the following, we address the weaknesses and the changes you have requested.
>
> > - The recency bias section, while conceptually interesting, feels underdeveloped. The explanation of why recency bias transitions from sharp to U-shaped is somewhat speculative.
> > - Consider including a deeper analysis or a concrete hypothesis to explain the emergence of the U-shaped perplexity profile during cooldown. This would sharpen the section’s contribution.
>
> In the paper, we state that "this can be explained by the fact that the final cooldown steps are performed with a near-zero learning rate, making the impact of the final data points negligible". While we provide an intuitive explanation of such behavior, this phenomenon was analyzed in-depth by Bergsma et al. (2025). We provide an additional reference near the U-shape behavior discussion.
>
> > - It is unclear whether the difference in validation perplexity between different cooldown shapes is significant enough to study the problem or not. If the difference is too negligible, people would just choose the random one to use.
> > - Would it be possible to include downstream task performance in the evaluation?
>
> We acknowledge that not providing downstream task performance is a limitation of our paper. This is due to the fact that to see interpretable results, we need to train large models with different schedulers, which is resource-consuming. In this regard, we rely on the findings of Hägele et al., 2024, who showed that performance improvement on validation from learning rate scheduler shape choice roughly maps to downstream task performance.
>
> > - Some of the figures, particularly in Section 4, are rich but a bit dense. For example, Figure 9 would benefit from more precise legends.
>
> Thanks for the suggestion! We have made text labels cleaner and ensured they do not overlap.
>
> > - The discussion on model averaging versus longer training stops short of offering a broader conclusion. It would be helpful to include a take-home message about when averaging is or isn't worthwhile.
>
> We include a take-home message, which is "it is also evident that high-variance low-bias shapes provide the best performance when performing averaging". The question of "should we perform averaging or not" is covered by related papers and usually emerges from the need to combine different high-quality datasets in one model or as a method of obtaining a universal model from several expert models. Therefore, this question falls outside the scope of our paper.
>
> > - The notion of recency bias is intriguing, but currently, the conclusions feel tentative.
>
> We use the recency bias perspective as one of the possible explanations for the emerging trade-offs. Also, recency bias provides a more practical perspective, as bias towards recent points is easier to understand and use in practice than distances in arbitrary spaces (bias-variance plots).

---

### Review · Reviewer_4hbb · 2025-05-28

**Summary Of Contributions:**

This article studies how learning rate scheduling affects the performance of large language models. By focusing on the cooldown phase to fine-tune a pretrained model, a bias-variance trade-off analysis is given to show that the optimal performance is often achieved when a bias-variance balance is achieved. This phenomenon is observed based on various decay scheduling including warmup-stable scheduler. The impact of batch size as well as AdamW hyperparameter tuning is also studied, showing the importance of optimization in this training phase. From a loss landscape perspective, the article finds a river valley shape validation loss during the cooldown phase.

**Audience:**

Yes

**Claims And Evidence:**

No

**Requested Changes:**

- Add further theoretically justification of the proposed framework.
- Rewrite eq (1), make the definition of each term clearer. Is k the index a sequence of length K?
- A definition of optimal bias/variance trade-off is needed to understand better the results in Figure 7 and 8.
- eq (2) is very hard to understand. What does that mean that “i in cooldown”? This is not mathematically meaningful.
- The loss landscape in Figure 13 could be made clearer to illustrate its connection with the idea given in Figure 1.

**Strengths And Weaknesses:**

The article gives a good overview of the learning rate scheduling literature, as well as extensive numerical results, leading to interesting insights and practical guidelines on how to train LMM. Furthermore, a bias-variance framework is proposed to analyze the validation perplexities of LMM, aiming to correlate the improved performance with proposed bias/variance quantities.

The proposed bias and variance definitions are, however, not so clear. It is hard to understand the central equation (1), where the definition of K is not clear. The norm with axis=k is not a common mathematical definition, therefore it needs to be clarified. Conceptually, there is a lack of connection between the bias and variance, as the bias is based on KL divergence, while the variance is based on MSE. From this perspective, the proposed framework is theoretically insufficient, and further explanation of the proposed definitions is needed.

It seems that there is no direct connection between the AdamW part (section 6) with the bias-variance trade-off. Therefore, the contributions seems to be dis-connected.

---

> ### Author Response · Authors · 2025-06-12
> **Implemented Changes**
>
> Thank you for the feedback!
>
> In the following, we address the weaknesses and the changes you have requested.
>
> > - The proposed bias and variance definitions are, however, not so clear. It is hard to understand the central equation (1), where the definition of K is not clear. The norm with axis=k is not a common mathematical definition, therefore it needs to be clarified. Conceptually, there is a lack of connection between the bias and variance, as the bias is based on KL divergence, while the variance is based on MSE. From this perspective, the proposed framework is theoretically insufficient, and further explanation of the proposed definitions is needed.
> > - Add further theoretically justification of the proposed framework.
>
> Thanks for the suggestion regarding Equation (1). The addition of "axis=k" should have made the formula easier to understand. We have removed the mention of "axis=k" in favor of standard mathematical notations and fixed the averaging issue. In addition, we added a definition of K.
>
> We moved this definition of the bias-variance plot to the appendix and replaced it with a new, easier-defined and more intuitive definition. We also discuss this change in a changes overview post.
>
> > - It seems that there is no direct connection between the AdamW part (section 6) with the bias-variance trade-off. Therefore, the contributions seems to be dis-connected.
>
> We are sorry to see that the connection is not noticeable. We present our incentive in the introduction of Section 6, which is to analyze how optimizer's hyperparameter modifications compare with cooldown shape choice. Our findings demonstrate that the impact of changes to Adam betas is comparable to the choice of cooldown shape in terms of validation performance. In addition, in Figure 11 we show that the relative performance of the cooldown shapes remains unchanged within a reasonable range of beta values.
>
> Perhaps, if you could clarify which specific connection is not persuasive, we could explain it further.
>
> > - A definition of optimal bias/variance trade-off is needed to understand better the results in Figure 7 and 8.
>
> The optimal cooldown shape from the bias/variance trade-off is whatever results in the lowest bias + variance in the setup. We decompose the loss into two sources of error that contribute to a higher loss. Generally, we would like to minimize both, but since there seems to be a trade-off (empirically), we just have to choose what gives us the minimal bias + variance. We added such clarification to the main text.
>
> > - eq (2) is very hard to understand. What does that mean that “i in cooldown”? This is not mathematically meaningful.
>
> We clarified the equation with standard mathematical terms and have updated the paper. Specifically, we introduced the set of batch indices that belong to the cooldown stage in Equation (2).
>
> > - The loss landscape in Figure 13 could be made clearer to illustrate its connection with the idea given in Figure 1.
>
> Thanks for the suggestion. For loss landscape plots, we do not see how we can make a visual connection to Figure 1. At this point, loss landscape plots serve as empirical visualizations of the river valley descent perspective. We added clarification to the text, specifying that from the "river-valley" perspective, progress downstream along the river corresponds to minimizing the bias, getting closer to the bottom of the valley is minimizing the variance. The fact that a local optimization step (from a single example) is almost orthogonal to the best global direction is exactly what causes the variance; we can't make progress downstream (reducing the bias) without taking large steps along the orthogonal directions (causing variance).

---

### Review · Reviewer_ooiE · 2025-05-29

**Summary Of Contributions:**

This paper is a collection of loosely connected investigations on the training dynamics of the cool down stage of pre-training a language model. The investigations can be roughly divided into 4 parts:

1. Trying different decaying schedules of the learning-rate, and connecting the performance to a bias-variance trade-off perspective. Among usual schedules such as linear, cosine, sqrt, etc., the authors also tried a family of "lowered linear" schedules, and show that the family demonstrates a negative correlation curve on the bias-variance plot. Moreover, this family appears to be on a Pareto front of the bias-variance plot, among all schedules being investigated. It suggests that there is indeed a trade-off between the bias and variance defined in the paper, which might control the training performance.

2. Comparing model averaging with longer training.

3. Investigating other Adam hyper-parameters such as batch size, beta1 and beta2 for the cool down stage. Especially, the finding that larger beta2 at the cool down stage can improve performance, might be useful empirically for future practice and study.

4. A nice visualization of the loss landscape, clearly showing a "river valley" like structure, which supports previous conjectures.

**Audience:**

Yes

**Broader Impact Concerns:**

None.

**Claims And Evidence:**

Yes

**Requested Changes:**

I don't want to block this paper, because it contains quite a lot of meaningful, inspiring findings; but it would definitely be better if the authors can establish more theoretical justifications of the bias-variance trade-off, such as a formal bias-variance decomposition, and more discussions on the potential logical connections between the label distribution space, model weights space, and the loss space -- to me, the similar bias-variance structure observed in the model weights space is not "surprising"; rather, it looks like a cause of the observed phenomena.

**Strengths And Weaknesses:**

All the experiments are meaningful, rigorous, and inspiring. The authors have good intuition on what might be happening during training, are knowledgeable in the literature, and have conducted concrete experiments for proof.

A high-level weakness is that the paper lacks a clear, overall story that can closely connect the 4 parts all together. Specific to each part, I would like to have some further discussions:

1. One big weakness with the current "bias-variance trade-off" perspective, is that the paper lacks an explicit bias-variance decomposition. Without such a decomposition, the definitions of bias and variance in the paper can be arbitrary (although, I understand the intuition and the definitions seem reasonable to me) but not well justified. There might indeed be a trade-off between "bias" and "variance", but it could also be due to some artificial math trick hidden in the definitions which make it look like so -- especially, when we look at similar bias-variance plots in the model weights space, as shown in Section C.1, the lowered linear family is no longer on a Pareto front -- which suggests that things can clearly change when the definitions of bias and variance are different.

2. Comparing model averaging with longer training: I was a bit confused when reading this section, because Figure 10 shows that averaged models consistently perform better than single runs, but the narration says "all averages perform worse than a simple longer run..."; then I realized that the paper is talking about the 52.8k sqrt run shown as a dashed line in Figure 10 (which is not very intuitive)...

3. When the hyper-parameters are changed, what are the error bars of validation perplexity, due to different data shuffles?

4. The river valley structure could also be explained as: because you are connecting the pre-cooldown checkpoint to the final model after cooldown as the "global optimization direction", at the end of the cooldown that direction is going to reach some valley because the training converges. On the other hand, a local optimization step (or 10 optimization steps) is likely orthogonal to the global direction because the parameter space is very high-dimensional. So what is special about this "river valley" perspective?

---

> ### Author Response · Authors · 2025-06-12
> **Implemented Changes**
>
> Thank you for the feedback!
>
> In the following, we address the weaknesses and the changes you have requested.
>
> > - A high-level weakness is that the paper lacks a clear, overall story that can closely connect the 4 parts all together. Specific to each part, I would like to have some further discussions:
> > - One big weakness with the current "bias-variance trade-off" perspective, is that the paper lacks an explicit bias-variance decomposition. Without such a decomposition, the definitions of bias and variance in the paper can be arbitrary (although, I understand the intuition and the definitions seem reasonable to me) but not well justified. There might indeed be a trade-off between "bias" and "variance", but it could also be due to some artificial math trick hidden in the definitions which make it look like so -- especially, when we look at similar bias-variance plots in the model weights space, as shown in Section C.1, the lowered linear family is no longer on a Pareto front -- which suggests that things can clearly change when the definitions of bias and variance are different.
> > - I don't want to block this paper, because it contains quite a lot of meaningful, inspiring findings; but it would definitely be better if the authors can establish more theoretical justifications of the bias-variance trade-off, such as a formal bias-variance decomposition, and more discussions on the potential logical connections between the label distribution space, model weights space, and the loss space -- to me, the similar bias-variance structure observed in the model weights space is not "surprising"; rather, it looks like a cause of the observed phenomena.
>
> Thanks for your suggestion. We conducted additional plots and discovered an easier-to-interpret bias-variance coordinate system. We included it in the main text and moved the previous bias-variance plot to the appendix. It is true that with the choice of bias-variance coordinates, the final plots may change. We believe that the new system is well-interpretable, intuitive, and aligns with empirical observations.
>
> > - Comparing model averaging with longer training: I was a bit confused when reading this section, because Figure 10 shows that averaged models consistently perform better than single runs, but the narration says "all averages perform worse than a simple longer run..."; then I realized that the paper is talking about the 52.8k sqrt run shown as a dashed line in Figure 10 (which is not very intuitive)
>
> We reviewed and clarified the text of Section 5 for clarity.
>
> > - When the hyper-parameters are changed, what are the error bars of validation perplexity, due to different data shuffles?
>
> We are preparing error bars for Figure 10 (different data orderings), but were not able to finish the experiments before the response deadline. We hope to update our submission as soon as the results are available.
>
> > - The river valley structure could also be explained as: because you are connecting the pre-cooldown checkpoint to the final model after cooldown as the "global optimization direction", at the end of the cooldown that direction is going to reach some valley because the training converges. On the other hand, a local optimization step (or 10 optimization steps) is likely orthogonal to the global direction because the parameter space is very high-dimensional. So what is special about this "river valley" perspective?
>
> We added clarification to the text, specifying that from the "river-valley" perspective, progress downstream along the river corresponds to minimizing the bias, getting closer to the bottom of the valley is minimizing the variance. The fact that a local optimization step (from a single example) is almost orthogonal to the best global direction is exactly what causes the variance; we can't make progress downstream (reducing the bias) without taking large steps along the orthogonal directions (causing variance).

---

### Author Response · Authors · 2025-06-12
**Review Responses and Changes Summary**

We thank the reviewers for providing feedback and suggestions. We responded specifically to each reviewer in the appropriate thread. Here, we summarize the main changes to the paper.

Several reviewers raised concerns about the interpretability and theoretical foundation of the chosen bias-variance coordinate system. We conducted additional experiments with different coordinate systems and found a new one that we believe to be well-interpretable, intuitive, and aligned with empirical observations. Therefore, the main text and plots concerning the bias-variance plot were adjusted to align with the new coordinate system.

We are preparing error bars for Figure 10 (different data orderings), but were not able to finish the experiments before the response deadline.

---

> ### Author Response · Authors · 2025-06-13
> **Section 6 Updates**
>
> Dear reviewers, we have now uploaded a new version with updates to Section 6. Specifically, error bars have been added to Figure 10 as requested by the reviewers. We have also added additional details about learning rate selection for the experiments in Section 6.1. Lastly, we noticed an inconsistency between the left and right plots of Figure 10 — the right plot used different base AdamW betas. We have fixed this mistake and updated the figure.

---

### Decision · Action_Editor_cafb · 2025-07-28

**Recommendation:** Accept as is

**Additional Comments:**

This paper explores various learning rate decay schedules and their impact on model performance from a bias-variance perspective. It also examines the relationship between Adam's hyperparameters and the cool-down phase, offering insights that are valuable for practitioners. The empirical results are meaningful and compelling, highlighting a strong connection to the theoretical analysis.

**Audience:**

Yes

**Audience Explanation:**

I believe a substantial portion of TMLR’s audience will find this new analysis valuable for training complex and large-scale models.

**Claims And Evidence:**

Yes

**Claims Explanation:**

This paper investigates the mechanisms underlying the cool-down phase in learning rate scheduling. Their analysis reveals that different cool-down strategies have distinct impacts on the bias-variance trade-off.